# Morphology of Larger Salivary Glands in Peccaries (*Pecari tajacu* Linnaeus, 1758)

**DOI:** 10.3390/ani14192891

**Published:** 2024-10-08

**Authors:** Carlos Magno Oliveira Júnior, Hélio Noberto de Araújo Júnior, Moisés Dantas Tertulino, Ricardo Romão Guerra, Luciana Diniz Rola, Alexandre Rodrigues da Silva, Carlos Eduardo Bezerra de Moura, Moacir Franco de Oliveira

**Affiliations:** 1Instituto Federal de Educação, Ciência e Tecnologia do Ceará, Campus Fortaleza, Fortaleza 60115-222, Brazil; carlosmagno@ifce.edu.br (C.M.O.J.); moises.tertulino@hotmail.com (M.D.T.); 2Programa de Pós-Graduação em Ciência Animal, Universidade Estadual do Ceará, Fortaleza 60714-903, Brazil; helio.noberto@uece.br; 3Programa de Pós-Graduação em Ciência Animal, Universidade Federal da Paraíba, Campus II, Areia 58397-000, Brazil; lucianadinizr@gmail.com; 4Programa de Pós-Graduação em Ciência Animal, Universidade Federal Rural do Semi-Árido, Mossoró 59625-900, Brazil; alexrs@ufersa.edu.br (A.R.d.S.); carlos.mora@ufersa.edu.br (C.E.B.d.M.); moacir@ufersa.edu.br (M.F.d.O.)

**Keywords:** wild animals, lecithins, *Tayassuidae*, ultrastructure

## Abstract

**Simple Summary:**

Peccaries are distributed from the southern United States to southern South America. Although they were initially described as being from the pig genus Sus, they are very different. This species is currently being used on commercial farms as a source of animal protein; however, there are no studies about their salivary glands, which are important for digestion and in developing feeding strategies. Thus, this work aims to study the major salivary gland morphology of peccaries during their growth. During growth, the parotid enlarges and the mandibular gland loses weight. Histologically, the parotid has serous production, and the sublingual has mucus production, like most species; however, the mandibular gland produces mucus, unlike other animals, including pigs, which produce seromucous secretion. Histochemically, the parotid produces more acidic mucins than pigs and it undergoes maturation during development; the mandibular, especially the sublingual gland, produces more acidic and basic mucopolysaccharides than pigs. The major salivary glands reacted positively to different lecithins and showed a stronger reaction than in pigs. We conclude that peccaries have a salivary secretion that facilitates the digestion of carbohydrates, improving digestibility and performance. Their biometric characteristics and positive reactions to lecithins facilitate adaptation to foods with antinutritional factors, making them a promising production animal.

**Abstract:**

This work aims to study the major salivary gland morphology of peccaries during their growth. The glands were analyzed using macroscopic description, light microscopy, electron microscopy, histochemistry, and immunohistochemistry. Topographically, the salivary glands resemble other animals, including domestic animals and pigs. During growth, the parotid enlarges and mandibular gland loses weight. Histologically, the parotid has serous production, and sublingual has mucous production, resembles most species, however, mandibular gland produces mucous, unlike other animals, including pigs, which produce seromucous secretion. Histochemically, parotid produces more acidic mucins than pigs and it undergoes maturation during development; mandibular, and especially the sublingual gland, produce more acidic and basic mucopolysaccharides than pigs. The results found with transmission and scanning electron microscopy techniques corroborate the histological and histochemistry findings. The major salivary glands were positive to different lecithins (Com-A, BSA-I-B_4_, WGA and PNA), which were also more positive than in pigs and sheep. We conclude that collared peccaries have a salivary secretion that facilitates the digestion of carbohydrates, and biometric characteristics and positivity to lecithins that facilitate adaptation to foods with antinutritional factors.

## 1. Introduction

The salivary glands are responsible for the synthesis and secretion of saliva directly into the oral cavity. They are divided by size, into larger and smaller ones. The largest salivary glands found in animals are parotid, mandibular, and sublingual. These glands can be defined according to the type of secretion released by their acini, in serous, mucous, or seromucous, which synthesize saliva through nervous coordination, increasing or decreasing salivary activity. The newly produced saliva by the acini is called primary saliva. As it passes through the salivary ducts, it undergoes ionic changes and variations in the amount of water before its secretion into the oral cavity [1].

In general, the major salivary glands, together with the minor ones, contribute to the maintenance of oral physiology, synthesizing and secreting saliva, moistening food and initiating the digestion of carbohydrates, assisting in oral lubrication, in taste, immunity, and oral homeostasis. It contains minerals, enzymes, and immunoglobulins, facilitating swallowing, chewing, vocalization, regulation of oral pH [2], oral protection and healing. Mucins such as MUC5b and MUC7, and proline-rich proteins in saliva are responsible for forming a protective film on teeth, preventing bacterial adhesion to the tooth [3]. Despite their importance, the morphology and histochemical characteristics of these glands are restricted to domestic animals such as pigs, sheep, cattle and horses, and practically non-existent in wild animals [4,5,6].

Studies suggest that salivary glands played a fundamental role in the terrestrial domination of mammals in various niches [7,8], contributing to evolution, as a result of modifying themselves to facilitate the digestion of the most diverse foods. These adjustments are due to acinar structures, made up of mucous and serous cells, each synthesizing and secreting a different granule. The secretory granules are released into the oral mucosa through the ducts. These granules can differ even between closely related species [9], and also in relation to different stages of maturation [10]. There are reports that salivary glands help species to adapt to diets containing antinutritional factors, such as tannin in pigs [2]. This domestic species, is comparable to the collared peccary (*Pecari tajacu*), but has different characteristics, including a stomach that is similar to that of ruminants [11]. Peccaries are distributed from the south of the United States to the Andes, reaching northern Argentina. They are artiodactyl mammals, from the *Tayassuidae* family, initially described by Linnaeus, in 1758, as belonging to the Sus genus [12]. The stomach of the collared peccary contains a glandular portion, and two blind sacs, one cranioventral and one caudo-dorsal [11], allowing for several items within their menu, including fibrous items [13].

Studies addressing the morphology of *Tayassuidae* do not involve information about parotid, mandibular and sublingual glands, which have great importance for digestion, adaptation, and homeostasis. These animals are being domesticated and used recently as a protein source in commercial farms in Brazil. Therefore, this work aims to fill the gap of information about morphology of larger salivary glands in collared peccaries at different periods of development, generating subsidies to the preservation of this species in wildlife screening centers, and to develop strategies for the formulation of diets and nutrition.

## 2. Material and Methods

### 2.1. Location and Collection of Material

The study used eight peccaries of different ages (two 4 months old, two 5 months old, two 6 months old, and two 7 months old) of both sexes, obtained from the Wild Animal Multiplication Center—CEMAS/UFERSA, a scientific breeding facility, registered at IBAMA under number 1478912. The animals were kept in paddocks, and the diet was grounded for pigs, with 18% protein based on wheat bran, corn and soybean. In addition, once a week complementary feeding was given with fruits, potatoes, carrots and pumpkins. Water was ad libitum.

### 2.2. Euthanasia Protocol

The animals were first subjected to pre-anesthesia with the use of xylazine (1 mg/kg) and ketamine (10 mg/kg) applied to the semi-membranous muscle. Anesthetic induction was performed intravenously (cephalic vein) with administration of thiopental (25 mg/kg). Then, a potassium chloride solution (1 mk/kg) was infused through the same venous access.

### 2.3. Macroscopic Analysis

After euthanasia, the animals were dissected in the animal anatomy laboratory. The parotid, mandibular and sublingual glands were weighed using a precision scale (AL200C, Marte, Rio de Janeiro, Brazil, 200 g ± 0.001 g), following Scherle’s methodology [14]. To evaluate length, width, and thickness, the glands were measured by using a digital caliper (Digimatic Caliper, Mitutoyo, Kawasaki, Japan, 150 mm). The length was measured based on the greatest extension of the gland, and the width based on its smallest extension. As the parotid gland had an irregular surface, when measuring width and thickness, three locations were measured, and an arithmetic average was taken to infer the measurement of the gland.

### 2.4. Microscopic Analysis

The sections were made randomly. Tissue fragments measuring 0.5 cm^3^ from the parotid, mandibular and sublingual glands were removed and fixed in 4% paraformaldehyde buffered with 0.1 M sodium phosphate, pH 7.4 at 4 °C.

After fixation, the material (major salivary glands) was washed in running water and subjected to the methodology adapted from Tolosa et al. [15] to obtain the slides. The slides with 05-μm-thick sections, 08 of each gland per animal, were stained with Hematoxylin-Eosin (HE); Periodic Acid-Schiff (PAS) to detect neutral mucins and labile sialomucins; Alcian Blue (pH 2.5) to detect sialomucins and sulfomucins; PAS associated with Alcian Blue (pH 2.5) to detect acini that synthesize certain mucins; and Gomori Trichrome for marking collagen fibers. For the PAS and Alcian Blue (pH 2.5) methodology, a control group known to be positive for these stains was used.

Their analysis was carried out by using a light microscope (LEICA ICC50W, Wetzlar, Germany) with magnification ranging from 10× to 100×.

### 2.5. Immunohistochemistry

For this step, the protocol adapted from Çinar, Öztop and Özkarasu [16] was used. The same biological samples used for the histology and histochemistry techniques were used in immunohistochemistry. 5-μm-thick sections of the parotid, mandibular, and sublingual glands were adhered to silanized slides, deparaffinized in xylene, rehydrated in decreasing concentrations of ethanol for 5 min, immersed in 3% hydrogen peroxide for 30 min, washed in 3 immersions of 5 min in distilled water, and then 3 immersions of 5 min in 0.1 mM phosphate buffer solution (PBS) in each wash.

Antigen exposure of the tissues was carried out in citrate buffer (10 mM, pH 6.0), conditioned in a microwave oven (60 °C for 15 min), followed by three washes in PBS. In the next step, endogenous peroxidase and nonspecific reactions were blocked by using the DAKO Kit in a humidity chamber for 30 min, followed by three 5-min washes in PBS and continuing with the incubation of the specimens with lecithin in a humidity chamber. Four lecithin were used: *Canavalia ensiformis*—Con-A (Reference: C2272, Sigma-Aldrich^®^, Rio de Janeiro, Brazil), *Bandeiraea simplicifolia*—BSA-I-B4 (Reference: L2140, Sigma-Aldrich^®^), *Triticum vulgaris*—WGA (Reference: L5142, Sigma-Aldrich^®^), *Arachis hypogaea*—PNA (Reference: L6135, Sigma-Aldrich^®^) (Table 1). The negative control was made by using PBS in a humidity chamber instead of lecithin for each tissue.

After an incubation of approximately 8 h, the lecithin was removed and washed three times for 5 min with PBS, followed by incubation in Spring Reveal Complement (Spring Bioscience^®^, Pleasanton, CA, USA) for 30 min, and then in Spring Reveal HRP Conjugate (Spring Bioscience^®^) also for 30 min. The development was performed with the chromogen Diaminobenzindine (Scytek DAB Sustrate Cromogen-Bioscience^®^, Lakeview Boulevard Fremont, CA, USA) diluted in a specific substrate (Scytek DAB Substrate-Bioscience^®^, Lakeview Boulevard Fremont, CA, USA). The slides were mounted by using Permount^®^ (Fisher Scientific, Waltham, MA, USA), then analyzed and photographed (Leica, ICC50W, Wetzlar, Germany).

The slide readings were performed by the same pathologist, analyzing the positivity for each antibody by the intensity of the DAB chromogen staining, in a qualitative manner, a technique commonly used in pathology.

### 2.6. Transmission Electron Microscopy

Three random portions of the parotid, mandibular, and sublingual major salivary glands were collected with dimensions of 0.5 mm^3^, and then immersed in a 2.5% glutaraldehyde solution buffered with 0.1 molar sodium phosphate at pH 7.4 at 4 °C for fixation. After fixation, the samples were washed in sodium phosphate buffer solution and then post-fixed in 2% osmium tetroxide and then contrasted with 3% uranyl acetate. Subsequently, the material was dehydrated in increasing concentrations of ethyl alcohol, washed with propylene oxide, propylene oxide resin, and Spurr resin. The polymerization of the resin was completed when the material was placed in an oven at 60 °C for 72 h. The blocks were cut with an ultramicrotome (Ultracut R, Leica Microsystems, Wetzlar, Germany) and then contrasted with 2% uranyl acetate and 5% lead citrate for subsequent visualization in a transmission electron microscope (Morgagni 268D, FEI Company, Eindhoven, The Netherlands; Mega View III camera, Soft Imaging, Münster, Germany), and the most representative regions were electron micrographed.

### 2.7. Scanning Electron Microscopy

After collection, the fragments of the parotid, mandibular, and sublingual glands of the specimens were fragmented or cryofractured with liquid nitrogen and fragments measuring 0.5–1 cm^3^ were obtained. Then the material was fixed in a 2.5% glutaraldehyde buffer solution with 0.1 M sodium phosphate, 7.4 pH at 4 °C. Once fixation was complete, the fragments were washed 3 times, 5 min each, in distilled water and then post-fixed with 0.5% osmium tetroxide buffer with 0.1 M sodium phosphate and a pH of 7.4, for 40 min. After the post-fixation, the material was washed 3 times for 10 min each in distilled water. The material was then dehydrated by immersing in different and ascending concentrations of alcohol.

Once dehydration was completed, the material was dried at the critical point (Quorum K 850, Laughton, UK) using carbon dioxide (CO_2_), affixed to a support (Stub) and metallized with gold by “sputtering” (Quorom Q 150R ES, Laughton, UK), for observation of the images in a scanning electron microscope (Tescan Vega, 3 LMU, Kohoutovice, Czech Republic).

## 3. Results and Discussion

### 3.1. Macroscopic Studies

#### 3.1.1. Parotid Gland

With a superficial dissection and removal of the parotidauricularis muscle, it was possible to observe the parotid gland (one on each side), which is the largest salivary gland, completely covering the mandibular gland, located caudal to the masseter muscle. It had a triangular shape, like the African squirrel’s (*Epixerus ebii*) [17], the South American raccoon’s (*Procyon cancrivorus*) [18], and the minipig’s [19]. Differently, the parotid of pumas (*Puma concolor*) has a semilunar shape [20]. The peccary’s parotid gland has external macro-lobation and a pale red color, unlike that found in puma which is yellowish gray [20].

The parotid was well adhered to the adjacent connective tissue and found in a depression close to the ear (Figure 1). It had characteristics similar to those observed in the peccary (*Tayassu pecari*) [21] (other member of Tayassuidae family), domestic pigs, and in humans [22,23,24]. It was located on the lower maxillary ramus, below and in front of the external ear, as in wild dogs (*Cerdocyoun thous*) [20], and pumas [25]. It was surrounded by a fibroadipose capsule in a depression whose anterior limit is the masseter muscle; its upper limit is the zygomatic arch; and its lower limit is the anterior border of the sternocephalicus muscle. The ventral buccal branch of the facial nerve passed ventrally and rostrally to the parotid gland, and the dorsal portion of the facial nerve passes over the parotid gland (Figure 1).

The parenchyma was organized into lobes separated by dense connective tissue, lymph, nodules, and fat; macroscopic description similar to the parotid of the domestic pig [22], crab-eating raccoon [18], and squirrel [17].

The parotid duct of collared peccaries arises in the deep portion of the gland and in its course passes medially to the angle of the mandible, rostrally bypassing the masseter muscle, and immersing in its rostral margin. Next to the parotid duct, it was possible to verify the presence of the facial vein. The parotid duct opened into the maxilla, between the molar teeth.

In the present study unfortunately, we are unable to perform statistics on the size of the salivary glands. Since the animals in this study are from a wild species that will serve as breeding stock for the scientific and conservationist breeding facility at our University, we cannot euthanize a sufficient number to perform statistics within each age group (4, 5, 6 and 7 months). Therefore, the numbers demonstrate a trend. We observe an increase in the weight and size (width, thickness, and length) of the parotid gland is observed as the animal ages (Table 2), indicating that the function of this salivary gland could be important in the animal’s adult life.

In addition to being the largest salivary gland, the parotid gland is important for adapting to the consumption of certain diets, for example, when there is the inclusion of antinutritional factors such as tannin in the pig’s intake. In these cases, the parotid gland hypertrophies, proved to be the greatest defense mechanism in diets with a high tannin content. On the other hand, in these same cases, the mandibular gland decreases in size [2].

#### 3.1.2. Mandibular Gland

The mandibular gland was located close to the angle of the mandible. The mandibular is related rostrally to the mandibular lymph nodes, the sublingual gland and the masseter and digastric muscles; medially, to the digastric, the external carotid artery and the medial retropharyngeal lymph node; and caudally, to the geniohyoid muscle (Figure 1). In wild pigss, its position is slightly different, and is located in the atlantal fossa, extending to the basihyoid bone [25]. In minipigs, this gland is described superficially to the suprahyoid and infrahyoid muscles [19]. Its capsule continues rostrally with the monostomatic portion of the sublingual gland, in which it is firmly fused. Its shape is ovoid, one on each side of the collared peccary’s face, it is compact and with faded red color, unlike the grayish color seen in pumas [20]. In minipigs, it has been described as an inverted pear shape with a light-yellow color [19]. This gland is completely covered by the parotid gland, as in other species [19], and its duct starts out from its deepest portion and runs toward the oral cavity until it opens at the base of the tongue. During this route, the duct passes between the monostomatic and polystomatic sublingual glands. The linguofacial artery and vein provide its blood supply (Figure 1). This description resembles the mandibles of koalas (*Phascolarctos cinereus*) [26], crab-eating raccoons (*Procyon cancrivorous*) [18], and puma [20].

The mandibular duct of the collared peccary follows the same direction as the duct of the crab-eating raccoon. Furthermore, both in the collared peccary and in the crab-eating raccoon, the mandibular is oval, in a lobulation and anatomical position, differing only in relation to the coverage of the parotid, because in the crab-eating raccoon the mandibular is only partially covered by the most ventral part of the parotid [27].

Regarding the variation in mandibular weight, it was found that it decreases as the animal grows, while the width, thickness, and length of the gland increases as the animal ages (Table 3), as in pigs [2]. This gland must be more important during the fetal period.

#### 3.1.3. Sublingual Gland

The sublingual gland of peccaries is divided into two portions: monostomatic (caudal) and polystomatic (rostral). The polystomatic portion is larger than the monostomatic portion. The polystomatic portion is located on the bottom of the oral cavity, close to the mandibular symphysis. The cranial segment is close to the geniohyoid muscle, the back of the gland rests on the mylohyoid muscle, and ventrally the gland is syntopic with the hypoglossal muscle. This macroscopic appearance is similar to that observed in humans, dogs, cats, agoutis, and crab-eating raccoons [18,28]. The main duct of the mandibular gland passes through the ventral portion of the gland and several small ducts from the sublingual gland converge into the mandibular duct, as in pumas [20].

In the crab-eating raccoon [18] and the wild dog [25], the monostomatic portion is located in the occipito-mandibular region of the digastric muscle, while the polystomatic portion is located between the mucous membrane of the mouth and the mylohyoid muscle in crab-eating raccoons [18], and between the mucosa and the buccinator muscle in wild dogs [25]. Blood supply and venous drainage occur through branches of the lingual artery and vein, similar to humans [28].

The morphometric analysis of the polystomatic portion demonstrated that the weight, width, thickness, and length of the gland increased with the advancing age of the animal, and the right and left antimeres showed divergences (Table 4).

The monostomatic portion is smaller, located rostrally to the mandibular gland and its small size makes it look like a continuation of the mandibular gland. Differentiation is only possible through microscopy. Its shape resembles a bean, and its duct converges into the mandibular duct and from there into the oral cavity. The morphometric analysis of the monostomatic portion demonstrated that its weight, width, thickness, and length of the gland increased, with the advancing age of the animal. Furthermore, the right and left antimeres also showed divergences (Table 5).

### 3.2. Microscopic Studies

#### 3.2.1. Histology

##### Parotid Gland Microscopy

The parotid gland had a capsule of dense, non-modeled connective tissue, and from it delimiting lobes septa (Figure 2). These lobes had their own blood, lymphatic vessels, and nerves. The glands were of the compound tubuloacinar type, as in other species [23,29,30,31,32]. This is the gland that showed the most connective tissue, and a good amount of blood and ductal framework.

It is a strictly serous gland, made up of mostly pyramidal cells, with the apex oriented towards the acinar lumen, and basal nuclei, with chromatin clusters that are uniformly distributed, with a single nucleolus generally visible within each nucleus. The base of these cells was focused on the connective stroma, with all these characteristics corroborating the findings for other species, such as pigs [23], camels [33], sheep [31], newborn buffalos [32], Indian cattle (*Bos indicus*) [34], and African squirrels [17]. The histological description for the parotid gland in peccaries is also similar to that described for four species of marsupials of the genus *Macropodiae*, with regard to the lack of mucous cells, and the presence of vacuolated cytoplasm, indicating the presence of water or alcohol soluble material, as well as secretory granules irregularly dispersed at the ends of the secretory cells filling almost the entire cytoplasm, giving the acinar cell a vacuolated appearance [35]. However, wild buffaloes were described as having seromucous parotids [33], like some mucous cells found in carnivores [36].

Numerous secretory granules were arranged at the apex of the acinar cells with basophilic secretory granules, demonstrating high affinity for hematoxylin (Figure 2). The granules originate in cell synthesis processes and are secreted into the intercalated ducts, striated ducts, and excretory ducts, arranged in that order to carry saliva to the oral cavity. Goblet cells have been found in the epithelium of ducts, as in pigs [23]. External to the acini, myoepithelial cells capable of contraction were found, corroborating studies on other species [17,23].

Throughout glandular development, it remained serous, always containing serous acini, and its stroma was more complex than in other salivary glands. Their large size comes from the fact that they produce the digestive enzyme amylase, indicating the need to digest carbohydrates, since these animals also feed on fruits and nuts which are rich in carbohydrates. This enzyme also serves as a defense against pathogens in the oral cavity [17].

##### Mandibular Gland Microscopy

The glands are surrounded by a capsule of dense, non-shaped connective tissue, just like the parotid gland. From the capsule, connective tissue septa extended to the interior and subdivided the glandular parenchyma into numerous lobes with their own blood supply and nerves, in a similar configuration to the mandibular glands of gray short-tailed opossums (*Monodelphis domestica*) [37], crab-eating raccoons (*Procyon cancrivorous*) [27], minipigs [19], and koalas (*Phascolarctos cinereus*) [26]. Furthermore, in collared peccaries, connective tissue, rich in collagen, delimited the acini and ducts (Figure 3). Blood vessels and nerves are present in the connective tissue septa, as well as interlobular ducts.

The mandibular glands are mucous glands, thus differing from the koala glands which, according to Krause [26], are strictly serous. In minipigs and in domestic animals, the mandibular area is described as being seromucous [19]. Mucous cells are characterized by containing a large number of secretory granules in their cytoplasm that practically did not stain when using histological routine, however, they stained when using histochemical techniques. Mandibular secretion is more viscous than the saliva produced by the parotid, precisely because it is composed of mucous cells, which are rich in glycoproteins, including mucins [19]. It is mentioned that unlike the parotid, which is the main producer of saliva when stimulated, the mandibular is of great importance during sleep, when the parotid has no function [19].

The large production of secretory granules dilates the cytoplasm of acinar cells, compressing the nucleus and organelles of these cells in the basal region, flattening their nuclei. The morphological arrangement of the acinar cells was in accordance with that found in gray short-tailed opossums (*Monodelphis domestica*) [37], other species of marsupials [35], and koalas (*Phascolarctos cinereus*) [26]. No histological changes were observed throughout the glandular development studied.

##### Sublingual Gland Microscopy

Microscopically, both (monostomatic and polystomatic) sublingual glands are mucous (Figure 4 and Figure 5), differing from the sublingual glands of humans [38] and rats [39], which are seromucous. The strictly mucous composition of the peccary sublingual gland also differed from that of the adult rat sublingual gland [40], and also from the sublingual glands of agoutis (*Dasyprocta leporina*) [4], since two cell types with distinct characteristics are observed in these species.

The acini have mucus-producing cells characterized by having cytoplasm full of secretion granules; the nucleus was in the basal portion of the cells and its shape was flattened due to the large number of secretion granules. The mucous cells of the polystomatic sublingual gland resembled the mucous cells of the mandibular gland. Furthermore, the glandular parenchyma was divided by septa of dense non-modeled connective tissue, originating from the capsule (Figure 4 and Figure 5), like the other glands in the study. In the stroma and glandular parenchyma there are numerous ducts.

Its stroma is poorly developed; however, its acini are larger than the mandibular and parotid acini and are strongly eosinophilic. These morphological findings agree with those observed in four species of *Macropodiae* [35], differing in relation to the presence of serous cells.

##### Salivary Gland Duct Microscopy

In addition to the acini, the salivary glands had numerous ducts in their parenchyma that began small, permeating the acini and subsequently increasing and branching out. The smallest ducts were the intercalated ducts, followed by the striated ducts, from which emerged the excretory ducts that opened into the oral cavity (Figure 6).

The intercalated ducts permeated the acini and received secretory granules from the acinar cells. The cells of the intercalated ducts varied from simple squamous to simple cubic, with centrally located basophilic nuclei, and their cytoplasm was scarce and eosinophilic. This morphological arrangement was similar to what Krause [26] described in koalas (*Phascolarctos cinereus*), to what Mansouri and Atri [33] described in camels (*Camelus dromedarius*), and to what Boshell and Wilborn [22] described in domestic pigs. The intercalated ducts converged into the striated ducts. The latter contained mitochondria in their basal region and could have a single layer or two cell layers. The basal layer was cubic with a central basophilic nucleus, and then a layer of simple columnar cells with secretory granules in the apical portion. Their nuclei were basophilic and basal. Striated ducts receive this name due to the presence of striations in their basal region. These streaks are elongated mitochondria, suggesting intense cellular activity (Figure 6). The striated ducts of peccary resembled the striated ducts described by Krause [26] in koalas (*Phascolarctos cinereus*). The intercalated and striated ducts were found within the lobes and led to excretory ducts, which contained intralobular and interlobular portions and were covered by pseudostratified epithelium. The further it got from the acini, the more complex it became, and the larger its ductal lumen developed. The pseudostratified epithelium lost this characteristic and assumed a stratified form, having several layers of epithelial cells, becoming increasingly complex, culminating in the main duct that carries saliva to the oral cavity. Differently, in minipigs the interlobar duct is described as having tall prismatic cells [19]. Goblet cells were observed in the excretory ducts of the collared peccary. A similar description regarding the morphology and location of the excretory ducts in domestic pigs was presented by Boshell and Wilborn [22], and by Lentle et al. [35] in four species of wallabies (Marsupialia: Macropodiae).

### 3.3. Histochemistry

#### 3.3.1. Parotid Gland Histochemistry

Despite not having mucous cells, the parotid gland was lightly stained with Alcian Blue (pH 2.5) (AB) and also lightly stained with the periodic acid Schiff (PAS) method in specimens aged 4 and 5 months. From 6 months onwards, a transition was observed, where some specimens had their acini stained and others did not. From the age of 7 months on, the parotid no longer stained with PAS, making it possible to verify a change in the parotid salivary composition with the maturation of the gland. In the association of AB and PAS, the parotids revealed weak positivity for AB and imperceptible positivity for PAS (Appendix A). The weak histochemical positivity of the parotid showed that there is a discrete secretory activity of acidic mucins, resembling marsupials [35] and differing from domestic cattle [41]. These results also differ from those observed in domestic pigs [22]; while in pigs the majority of acinar cells ranged from PAS-negative to slightly PAS-positive, in collared peccaries above seven months of age, the parotid was always PAS-negative. Furthermore, domestic pigs contain negative AB (pH 2.6 or 0.5) acinar cells, therefore differing from the positive AB of collared peccaries. Thus, peccaries produce more acidic mucins in parotid saliva than pigs.

#### 3.3.2. Mandibular Gland Histochemistry

Histochemically, the mandibular gland was positive to AB (pH 2.5) (AB), to the PAS method, and to the association of PAS and AB, more strongly than the parotid (Appendix A), agreeing with what was found in minipigs [19] and supported by the fact of being a mucous gland, unlike the parotid, which is serous. It can be inferred that the synthesized mucins are neutral and also acidic. The positivity to the PAS method was similar to that described for kangaroos [35]. However, in these species (minipigs and kangaroos), the mandibular glands are seromucous, therefore also made up of PAS-positive mucous cells and PAS-negative serous cells, also substantiating findings in the mandibular gland of koalas [26] (Appendix A).

It should be noted that there is a greater secretory activity of acidic polysaccharides than in basic ones in the mandibular region, since there is an overlap between AB and PAS positivity.

#### 3.3.3. Sublingual Histochemistry

The sublingual glands were more positive for PAS, AB (pH 2.5) staining (AB) and also in the PAS and AB association, than the mandibular and parotid glands, reinforcing the hypothesis that they produce acidic and neutral mucopolysaccharides in large quantities (Appendix A). This histochemical structure differed from the sublingual glands of the tammar wallaby (*Macropus eugenii*), because the acini are weakly PAS-positive in this species [35].

In the sublingual glands, there are no histochemical differences between the two, and there is a greater secretory activity of acidic polysaccharides than basal ones, even when compared to the mandibular gland, since there is greater overlap of AB on PAS positivity.

### 3.4. Immunohistochemistry

#### 3.4.1. Parotid Gland Immunohistochemistry

Lectins are proteins or glycoproteins, which are of non-immune origin and are derived from plants, animals, or microorganisms that have specificity for terminal or subterminal carbohydrate residues. Lectin histochemistry can provide an extremely sensitive detection system for changes in glycosylation and carbohydrate expression that may occur during embryogenesis, growth, and disease. Lectin histochemistry can also reveal subtle alterations in glycosylation between otherwise indistinguishable cells [42]. Some studies have been carried out to determine the glycoconjugates found in parotid glands of some species by lectin histochemistry [16,43,44]. There are still no studies on lectins in peccaries, since this is the first study, and one of the first in wild species.

The parotid of the collared peccary was positive at varying intensities for the different glycoconjugates, depending on the lecithin used. These reactions also varied according to the age of the animals. There was no positivity in the parotid gland for glycoconjugates marked by the PNA lectin, both for four-month-old and seven-month-old animals, corroborating a study in sheep [16]. There was slight parotid positivity for BSA I-B4 and WGA, for both age groups, and moderate positivity for Con-A, also for animals of both age groups (Appendix A). With the exception of Con-A, which is moderately positive, the other results differ from those observed by Çinar, Öztop, and Özkarasu [16] when studying sheep, where the lecithin BSA I-B4, WGA, and PNA were also moderately positive.

Ten-day-old rats and newborn mice have mucus-producing cells in the parotid gland that are positive for BSA, WGA, and even PNA (not positive in peccaries). After 10 days of age, such cells become serous [45]. Possibly the difference between PNA positivity in newborn rats and mice and peccaries is due precisely to the fact that mucus-producing cells have changed into serosa-producing cells in rats and mice. In collared peccaries there is no evidence that parotid serous cells were ever mucous. Humans, Malayan pangolins, and lesser mouse deer also have their parotids positive to the same lecithin as peccaries, and negative to PNA [46,47]. Also, unlike peccaries, in horses, parotid serous cells are not BSA positive [48].

There are not many studies involving immunohistochemistry techniques for saliva components. Although there is a noteworthy study in pigs demonstrating that when antinutritional factors are included in the diet, positivity to basic lecithin increases, with no change in positivity to acid lecithin [2].

#### 3.4.2. Mandibular Gland Immunohistochemistry

In the collared peccary mandible, lecithin immunohistochemistry showed several positive reactions, which also varied according to the age of the animals. At four months, the mandibular test was strongly positive for PNA, BSA I-B4, and Con-A. However, at seven months, this positivity changes, going from strong to moderate for PNA and Con-A lecithin, and weak for lecithin BSA I-B4. WGA lecithin ranged from negative at four months to weakly positive at seven months (Appendix A).

#### 3.4.3. Sublingual Glands Immunohistochemistry

The immunohistochemical reaction in the peccary sublingual glands also varied according to the age of the animals. At four months and seven months of age, polystomatic sublingual is moderately positive for PNA, BSA I-B4, and Con-A. For WGA lecithin, polystomatic sublingual is weakly positive at four months and moderately positive at seven months (Appendix A).

In the monostomatic sublingual gland, there was moderate positivity at four months for PNA lecithin and strong positivity at seven months. Moderate positivity at both four months and seven months for BSA I-B4 lecithin. Strong positivity for Con-A lecithin, both at four months and seven months. For WGA lecithin, sublingual was moderately positive at four months and weak at seven months (Appendix A).

### 3.5. Electron Microscopy

#### 3.5.1. Parotid Gland Electron Microscopy

The parotid gland parenchyma is formed by compound acinar glands, separated by connective tissue, with two excretory portions joined by ducts. (Appendix A). The shape of the acinar cell is similar to a pyramid; however, this shape was not always observed (Appendix A), which differed from the acinar cells found in camels (*Camelus dromedarius*) [33] and resembling the acinar cells of domestic pigs [22]. The cells of the secretory units are full of secretory granules of different sizes, which are quite electron-dense and increase as they move away from the nucleus, differing from the electron-lucent granules of mice [49] and domestic pigs [22]. Mature secretory granules are close to the edge of the cell membrane, their electron density and size are greater than in immature granules.

The cellular nucleus of the parotid acini of peccaries is spherical, located at the cell base, and it contained a prominent nucleolus. The arrangement of cellular organelles in collared peccaries was basically organized in the basal region. The rough endoplasmic reticulum (RER) was in moderate quantity located mainly basal and lateral to the nucleus and could be seen randomly between the secretory granules. Other observable organelles are free ribosomes, some randomly distributed mitochondria with flattened cristae and sparse Golgi complex, which were in a similar arrangement to those described in domestic pigs [22]. Acinar cells synthesized and secreted granules into the intercalated ducts. With the exception of microvilli, which were not seen in collared peccary acinar cells, cell shape and arrangement of nuclei were similar to those observed in camels (*Camelus dromedarius*) [33].

Myoepithelial cells that contain few mitochondria and RER are around the acini, similar to those found in pig [22].

#### 3.5.2. Mandibular Gland Electron Microscopy

The ultrastructure of the mandibular glandular parenchyma is also divided into lobes by connective tissue (Appendix A). In collared peccaries, the connective tissue septa, originating from the capsule, divided the gland into lobes, acini, smaller intralobular ducts, and larger interlobular excretory ducts. This conformation is in line with that observed in mandibular salivary glands of crab-eating raccoons (*Procyon cancrivorous*) [27] and minipigs [19].

The acinar cell does not have a fixed shape, it varies from cubic to pyramidal. Its cytoplasm has countless electro-lucent secretion granules in different shades, some of which contain an electron-dense point inside. Mucous cells have RER, a protruding Golgi complex, and few mitochondria in their cytoplasm, all located basally. This is due to the large quantity of secretion granules, distributed mainly in the apical portion. The nucleus is spherical, located in the basal region of the cell, and it has a prominent nucleolus (Appendix A). This description differed from the mixed mandibular glands in mice [42], in Korean striped field mice (*Apodemus agraius*) [10], in Korean spider shrews (*Sorex caecutiens*) [9], and in minipigs [19], precisely because these animals have serous and mucous cells in their acini, and the serous ones are electron-dense. However, the arrangement of the organelles is similar among these species [9,11], and also in camels (*Camelus dromedarius*) [33].

#### 3.5.3. Sublingual Glands Electron Microscopy

In both portions of the sublingual gland, there are acini made up only of mucous cells with cytoplasm with numerous secretion granules of low to moderate electron density, thus, slightly more electron-dense than the granules found in the mandibular gland. Cellular organelles are found in the basal region, close to the cell nucleus (Appendix A). The strictly mucous composition of the peccary sublingual gland differed from the adult sublingual gland of rats [34], and also from the sublingual glands of agoutis (*Dasyprocta leporina*) [4], because in these species, two cell types with distinct characteristics are observed, and the serous granules are electron-dense.

Although topographically the salivary glands of peccary resemble other animals, including domestic animals and pigs, some morphological differences were observed. The peccary’ parotid enlarges with age, which may be related to the high quantity of foods rich in anti-nutritional factors that these animals find and consume in the forests of South America, as would be fatal for domestic species. This statement is corroborated by the increase in the parotids of pigs when they are subjected to diets rich in antinutritional factors, such as tannin [2]. The gland produces the digestive enzyme amylase, indicating the need to digest carbohydrates, since these animals also feed on fruits and nuts rich in carbohydrates [17]. Decrease in the mandibular gland is also seen in pigs subjected to antinutritional factors [2]. During the growth of mandibular peccaries, this gland loses weight, which increases the possibility that peccaries will adapt to this type of diet with many antinutritional factors.

Histologically, the parotid, with serous production, resembles most species, including domestic animals, the same happens for the sublingual gland with mucous production. However, the mandibular gland produces mucous in peccary, unlike other animals, including pigs, which produce seromucous secretion. Such differences in the mandibular glands may also be due to the fact that this gland, unlike the parotid gland that produces saliva when stimulated, is important during sleep [19]. Since peccaries sleep little and are often seen feeding at night [10,11], this gland may have its weight reduced with age, as already mentioned.

Still regarding the secretion of the glands, it was observed that the parotid gland produces more acidic mucins than pigs (positive for Alcian Blue, while in pigs it is negative) and that it undergoes maturation during development; and that the mandibular glands, and especially the sublingual gland, produce more acidic and basic mucopolysaccharides (demonstrated by the high positivity to PAS, AB and PAS + AB) than pigs. These characteristics provide peccary with better digestion of fruits and nuts when compared to pigs, most similar animal domestic, thus improving the digestibility of food even if nutritionally poorer.

The results found with transmission and scanning electron microscopy techniques corroborate the histological and histochemistry findings, and this is the first study with these techniques in collared peccary salivary glands. This is also the first study to look at positive lecithins in the salivary glands of collared peccaries, which were also more positive than in pigs and sheep [2,16], which also facilitates carbohydrate digestion in collared peccaries. The higher lecithin-positivity also indicates an adaptation of the collared peccaries to the ingestion of foods with antinutritional factors, since pigs when subjected to these factors also have increased lecithin-positivity in the salivary glands [2].

In addition to the morphological characteristics mentioned in the larger salivary glands of collared peccaries that provide digestive advantages for carbohydrates when compared to pigs, they also have another advantage, the presence of a stomach with four compartments (gastric pouch, cranioventral blind sac, caudodorsal blind sac and direct compartment), similar to those of ruminants [11,12], lead these animals to be capable of to digest more fibrous foods more efficiently through fermentation in these compartments. Remembering that dietary fibre was once considered to be a negative factor for monogastric animals due to its potential adverse effects on digestibility and performance, but collared peccaries cannot be considered monogastric. Such characteristics point to the collared peccary as a promising production animal, as it can satisfactorily digest both structural and non-structural carbohydrates.

## 4. Conclusions

This is the most complete article in literature concerning the morphology of salivary glands in any species of wild animal. The collared peccary has a major salivary glands secretion that facilitates the digestion of carbohydrates, improving digestibility and performance; and biometric characteristics and positivity to lecithins that facilitate adaptation to foods with antinutritional factors, being a promising production animal. These results serve as subsidies for the development of diet and nutrition formulation strategies for the species in commercial breeding or in wildlife screening centers.

## Figures and Tables

**Figure 1 animals-14-02891-f001:**
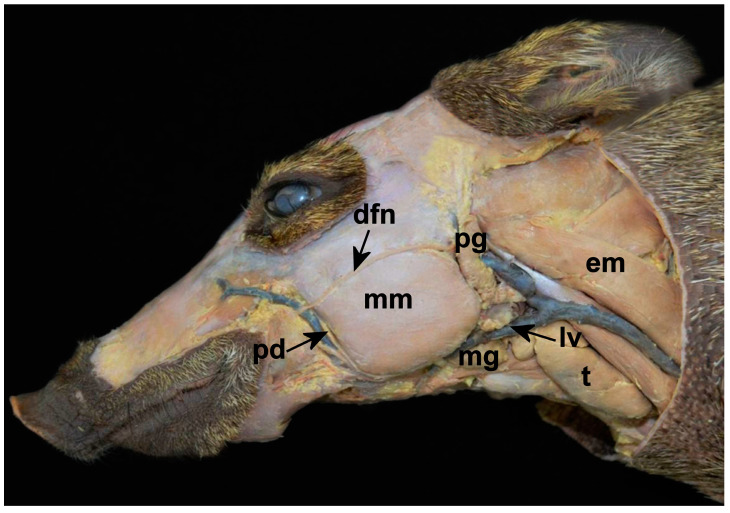
Photographic image of the macroscopy of the major salivary glands of the collared peccary (*Pecari tajacu* Linnaeus, 1758). Mandibular gland (mg) is located ventrally to the linguofacial vein (lv). Above is part of the parotid gland (pg), which is rostrally to the sternocephalicus muscle (em). The “em” is positioned dorsally to the thymus (t). Rostrally we still find the masseter muscle (mm), surrounded dorsally by the dorsal facial nerve (dfn), and caudally by the parotid duct (pd).

**Figure 2 animals-14-02891-f002:**
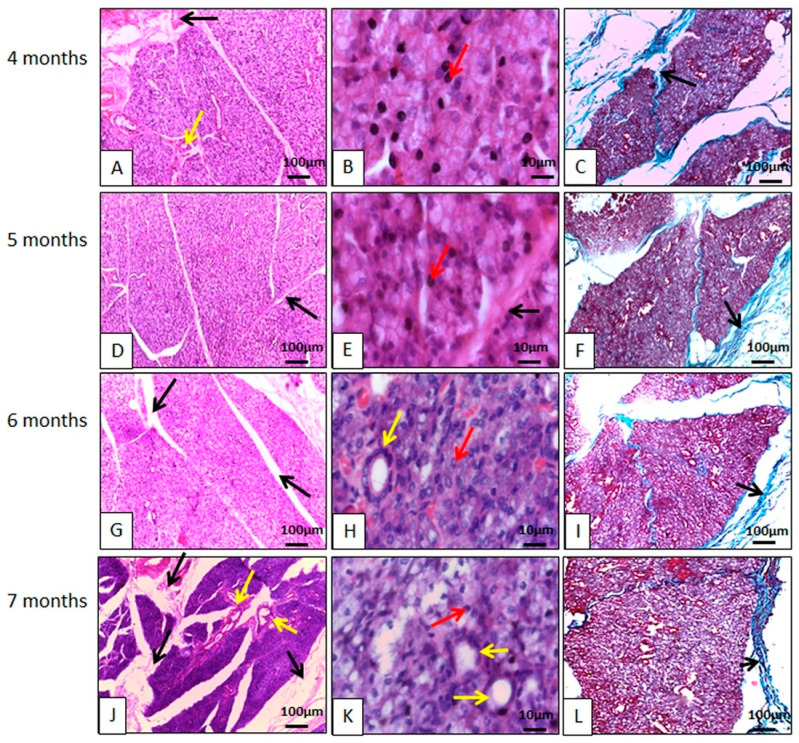
Photomicrographs of the parotid gland of the collared peccary (*Pecari tajacu* Linnaeus, 1758). In (**A**,**D**,**G**,**J**), it can be seen that the basophilic lobes are individualized by connective tissue (black arrows). In (**B**,**E**,**H**,**K**), the connective tissue (black arrows) delimits the acini and ducts (yellow arrows). The nuclei are spherical and are strongly stained by hematoxylin (red arrows). In (**C**,**F**,**I**,**L**), it can be seen that the lobes are individualized by collagen fibers (black arrows). (**A**,**B**,**D**,**E**,**G**,**H**,**J**,**K**) Hematoxylin-Eosin staining; (**C**,**F**,**I**,**L**) Gomori Trichrome Staining.

**Figure 3 animals-14-02891-f003:**
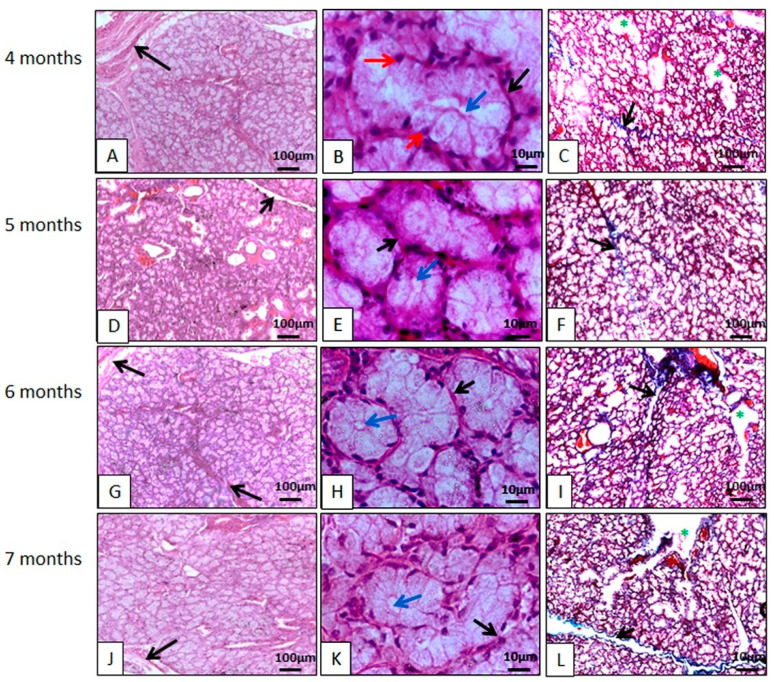
Photomicrographs of the collared peccary mandibular gland (*Pecari tajacu* Linnaeus, 1758). In (**A**,**D**,**G**,**J**), it can be seen that the acidophilic lobes are individualized by connective tissue (black arrows). In (**B**,**E**,**H**,**K**), connective tissue (black arrows) delimits the acini. The acinar cells are organized to form the acinar lumen (blue arrows). The nuclei of these cells are spherical and stain strongly with hematoxylin (red arrows). In (**C**,**F**,**I**,**L**), the ducts (*) are individualized by collagen fibers (black arrows). (**A**,**B**,**D**,**E**,**G**,**H**,**J**,**K**) Hematoxylin-Eosin staining; (**C**,**F**,**I**,**L**) Gomori Trichrome staining.

**Figure 4 animals-14-02891-f004:**
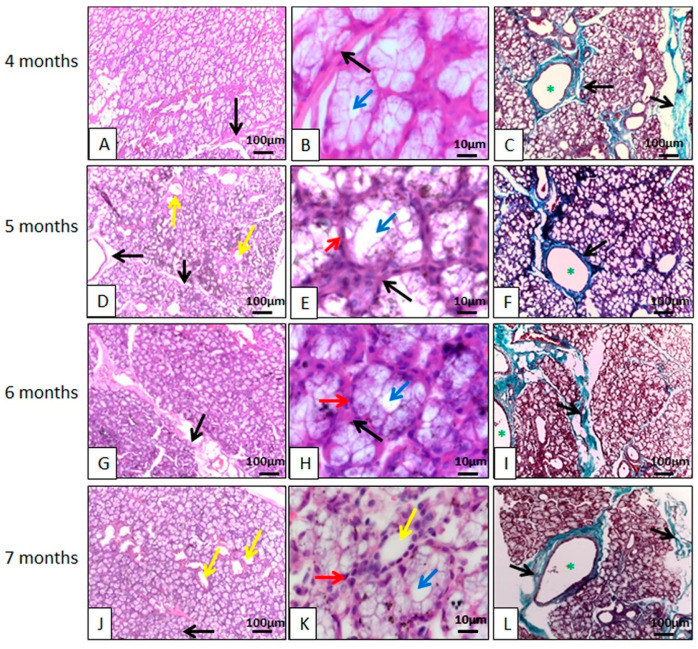
Photomicrographs of the polystomatic sublingual gland of the collared peccary (*Pecari tajacu* Linnaeus, 1758). In (**A**,**D**,**G**,**J**), the acini and ducts (yellow arrows) are individualized by connective tissue (black arrows). In (**B**,**E**,**H**,**K**), the acini and ducts (yellow arrows) are individualized by connective tissue (black arrows). The acinar cells are organized to form the acinar lumen (blue arrows), the nuclei of these cells are spherical and stain strongly with hematoxylin (red arrows). In (**C**,**F**,**I**,**L**), the connective tissue (black arrows) delimited the acini and ducts (*). (**A**,**B**,**D**,**E**,**G**,**H**,**J**,**K**) Hematoxylin-Eosin staining; (**C**,**F**,**I**,**L**) Gomori Trichrome staining.

**Figure 5 animals-14-02891-f005:**
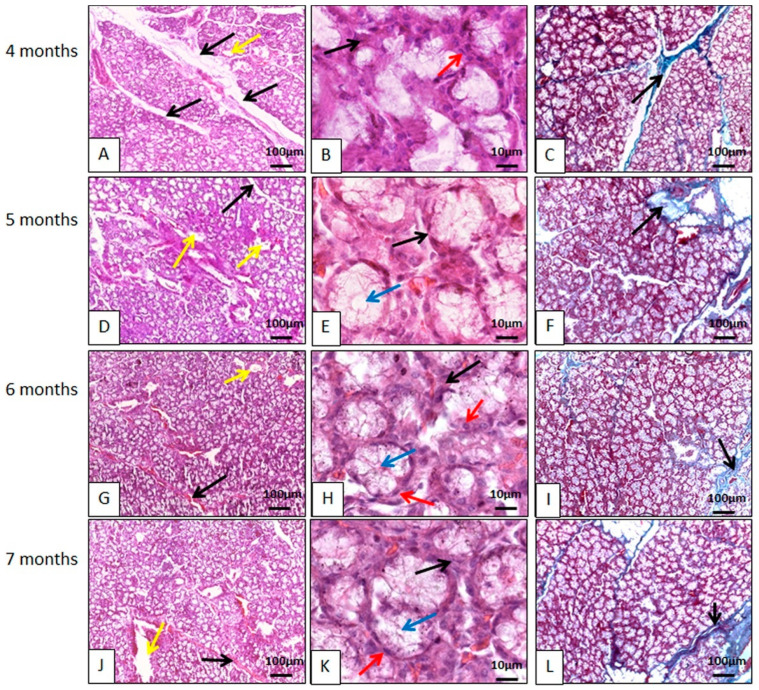
Photomicrographs of the monostomatic sublingual gland of the collared peccary (*Pecari tajacu* Linnaeus, 1758). In (**A**,**D**,**G**,**J**), the acini and ducts (yellow arrows) are individualized by connective tissue (black arrows). In (**B**,**E**,**H**,**K**) connective tissue (black arrows) delimit the acini and ducts (yellow arrows), the nuclei are flat, stain strongly with hematoxylin, and are in the basal region (red arrows). The acinar cells are with its apex facing the acinar lumen (blue arrows). In (**C**,**F**,**I**,**L**), the acini and ducts are individualized by collagen fiber meshes. (**A**,**B**,**D**,**E**,**G**,**H**,**J**,**K**) Hematoxylin-Eosin staining; (**C**,**F**,**I**,**L**) Gomori Trichrome staining.

**Figure 6 animals-14-02891-f006:**
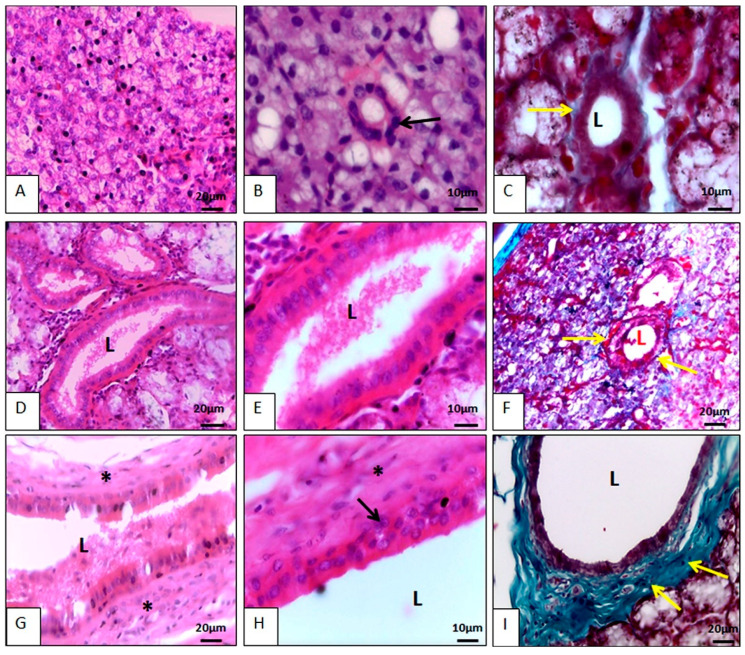
Photomicrographs of the salivary ducts of the collared peccary (*Pecari tajacu* Linnaeus, 1758). In (**A**), the arrangement of the intercalated ducts is observed. There is an intimate relationship between this duct and the acini. In (**B**), it can be seen that the duct is made up of a single layer of cubic epithelial cells (black arrows) and its nucleus is spherical, basophilic, and located in the cell center. In (**C**), the duct is delimited by a thin layer of collagen fiber (yellow arrows). In (**D**), there is the striated duct that was larger than the intercalated duct. In (**E**), it is observed that the cells were columnar, and the nuclei were spherical and positioned in the center of the cell. Note that the cell base is more eosinophilic than the cell apex. In (**F**), a thin layer of collagen fiber (yellow arrows) can be seen delimiting the duct. In (**G**), the excretory duct can be seen. This duct had a lot of connective tissue around it (*) and was in the interlobular region. In (**H**), it is noted that the excretory duct contained more than one layer of epithelial cells, the most basal of which was cubic (black arrow) and the outermost columnar. In (**I**), a dense network of collagen fibers is noted surrounding the excretory duct. L = lumen. (**A**,**B**,**D**,**E**,**G**,**H**) Hematoxylin-Eosin staining. (**C**,**F**,**I**) Gomori Trichrome staining.

**Table 1 animals-14-02891-t001:** Lecithins and concentrations used for immunohistochemistry of collared peccary (*Pecari tajacu*) major salivary glands.

Lecithins	Abbreviation	Carbohydrate Binding Specificity	Concentration (mg/mL)
*Canavalia ensiformis*	Con-A	αMan > αGlc > αGlcNAc	0.29 mg/mL
*Bandeiraea simplicifolia*	BSA-I-B_4_	αGal > αGalNAc	0.5 mg/mL
*Triticum vulgaris*	WGA	GlcNAc(β1,4GlcNAc)_1–2_ > βGlcNAc > Neu5Ac	0.1 mg/mL
*Arachis hypogaea*	PNA	Galß1,3GalNAc > α and βGal	0.1 mg/mL

**Table 2 animals-14-02891-t002:** Morphometric measurements and weight of the parotid glands of the collared peccary (*Pecari tajacu* Linnaeus, 1758) at 4, 5, 6, and 7 months of age.

Months	Parotid							
	Right Antimere				Left Antimere			
	L	W	T	WE	L	W	T	WE
**4M**	45.95	9.70	2.84	3.2396	45.01	10.30	2.77	2.8311
**4F**	46.11	16.37	2.19	3.6318	45.16	16.54	2.67	3.6437
**5F**	41.43	18.67	3.08	5.7121	42.97	13.70	3.48	5.0375
**5M**	43.03	16.97	5.75	5.7953	43.65	15.86	3.85	5.1178
**6F**	46.24	18.95	7.80	5.6066	47.93	16.70	6.24	5.4242
**6M**	53.22	14.79	6.32	5.1561	48.70	21.07	6.53	5.2172
**7F**	55.82	15.76	6.47	6.1577	48.70	21.07	6.53	5.8981
**7M**	56.95	12.52	9.53	6.1032	58.01	10.30	9.77	6.2119

L: length (mm); W: width (mm); T: thickness (mm) and WE: weight (grams). Female (F) and male (M).

**Table 3 animals-14-02891-t003:** Morphometric measurements and weight of the mandibular glands of the collared peccary (*Pecari tajacu* Linnaeus, 1758) at 4, 5, 6, and 7 months of age.

Months	Mandibular							
	Right Antimere				Left Antimere			
	L	W	T	WE	L	W	T	WE
**4F**	16.85	23.60	7.12	3.6871	17.22	20.13	6.77	3.2964
**4M**	14.24	25.72	8.38	3.8879	17.90	23.70	6.02	3.9315
**5F**	15.10	24.25	7.01	2.7472	15.47	26.46	7.21	2.8340
**5M**	14.76	21.29	6.74	3.0371	17.12	22.99	7.60	3.1759
**6F**	25.46	17.22	6.67	2.4162	24.90	16.44	8.48	2.3522
**6M**	20.34	27.02	8.85	2.8896	18.09	23.38	8.11	3.3432
**7F**	22.53	29.21	9.85	2.5533	20.28	27.27	9.30	2.0701
**7M**	20.17	27.02	8.85	2.7839	18.09	23.38	8.46	2.9218

L: length (mm); W: width (mm); T: thickness (mm) and WE: weight (grams). Female (F) and male (M).

**Table 4 animals-14-02891-t004:** Morphometric measurements and weight of the polystomatic sublingual glands of the collared peccary (*Pecari tajacu* Linnaeus, 1758) at 4, 5, 6, and 7 months of age.

Months	Polystomatic Sublingual							
	Right Antimere				Left Antimere			
	L	W	T	WE	L	W	T	WE
**4F**	35.08	5.19	2.80	0.8186	32.33	6.44	2.55	0.6738
**4M**	34.78	7.67	3.26	0.8671	26.82	6.37	2.83	0.9401
**5F**	34.62	6.33	2.29	0.8056	33.13	6.62	2.48	0.6538
**5M**	23.90	7.84	2.58	0.8396	22.01	8.52	2.64	0.9265
**6F**	33.30	7.43	3.74	1.1818	32.88	7.15	3.83	1.0788
**6M**	28.52	9.04	4.23	0.8511	29.58	10.05	3.65	0.7884
**7F**	31.52	8.54	4.19	1.5468	32.55	10.11	3.44	1.7033
**7M**	32.27	9.34	3.97	1.2518	31.44	9.75	3.88	1.3398

L: length (mm); W: width (mm); T: thickness (mm) and WE: weight (grams). Female (F) and male (M).

**Table 5 animals-14-02891-t005:** Morphometric measurements and weight of the monostomatic sublingual glands of the collared peccary (*Pecari tajacu* Linnaeus, 1758) at 4, 5, 6, and 7 months of age.

Months	Monostomatic Sublingual							
	Right Antimere				Left Antimere			
	L	W	T	WE	L	W	T	WE
**4M**	12.30	8.84	2.13	0.4833	12.01	7.47	2.20	0.2764
**4F**	12.25	4.91	1.27	0.3826	12.40	5.17	1.68	0.2774
**5F**	16.98	7.60	3.74	0.2685	15.64	8.78	3.54	0.4397
**5M**	10.62	7.18	3.61	0.4676	11.96	6.45	4.16	0.3986
**6F**	14.14	7.61	3.06	0.3211	13.40	9.88	3.52	0.5843
**6M**	17.78	8.45	3.47	0.4725	16.34	8.76	3.11	0.5663
**7F**	17.96	9.37	3.69	0.9533	17.70	9.78	3.11	0.7434
**7M**	17.38	9.48	3.47	0.7206	18.36	9.45	3.66	0.6892

L: length (mm); W: width (mm); T: thickness (mm) and WE: weight (grams). Female (F) and male (M).

## Data Availability

The data supporting the findings of this study are available from the corresponding author upon reasonable request.

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
