# Peer review of "Morphology of Larger Salivary Glands in Peccaries (Pecari tajacu Linnaeus, 1758)"

_animals, 2024, doi:10.3390/ani14192891_

Round 1
Reviewer 1 Report
Comments and Suggestions for Authors
Line 67 to 68: The sentence is repeated
Line 97: You should add a full stop between Brazil and this
Line 115: Where the animals embalmed for dissection?
Line 148 (Immunohistochemical techniques): Were all histological techniques performed on the same samples?
Line 222: You write "parotid auricular muscle" and you should name it "parotidoauricularis" or "parotideo-auricularis"
Line 235: You write "sternocleidomastoid muscle" and you should name it "sternocephalicus" to the mastoid process.
Line 235 and 236: You write "The buccal branch of the facial nerve passed ventrally to the parotid gland". You should say: "The ventral buccal branch of the facial nerve passed ventrally and rostrally to the parotid gland"
Line 241: The black arrow points at the dorsal buccal branch of the facial nerve not at the ventral. You have not specify if it is dorsal or ventral. However, in the text you seem to refer to the ventral.
Line 242: The same applies to the "sternocleidomastoid muscle"
Have you study the sympathetic and parasympathetic innervation of the salivary glands?
The histology images are a bit blur. May be is the quality of the pdf. You should check it.
Author Response
Referee I
Line 67 to 68: The sentence is repeated
Answer: The sentence repeated was removed: “Studies suggest that salivary glands played a fundamental role in the terrestrial domination of mammals in various niches [8,9], contributing to evolution, as a result of modifying themselves to facilitate the digestion of the most diverse foods”
Line 97: You should add a full stop between Brazil and this
A: The entire sentence has been changed, making the requested suggestion no longer possible. The sentence now reads as follows: “The importance of the salivary glands for digestion, adaptation, and homeostasis, and the anatomical differences in the digestive system of collared peccaries and pigs allows the first to digest even fibrous materials, increasing nutritional possibilities. Studies addressing the morphology of tayassuidaes do not involve information about the morphology of the parotid, mandibular and sublingual glands. These animals are being domesticated and used recently as a protein source in commercial farms in Brazil. Taking all these aspects into consideration, this work aims to fill the gap of information about such glands in collared peccaries at different periods of development, in order to support clinical and surgical knowledge, preservation of this species, and, to support strategies for the formulation of diets and nutrition, which nowadays receives nutrition similar to that of pigs.” (Lines 79-88)
Line 115: Where the animals embalmed for dissection?
A: In the line 112 was added: After euthanasia, the animals were dissected in the animal anatomy laboratory. (line 105)
Line 148 (Immunohistochemical techniques): Were all histological techniques performed on the same samples?
A: In the line 147-148 was added the sentence: “The same biological samples used for the histology and histochemistry techniques were used in immunohistochemistry” (line 130-131)
Line 222: You write "parotid auricular muscle" and you should name it "parotidoauricularis" or "parotideo-auricularis"
A: In the line 221, “parotid auricular” muscle was replacet to “parotidoauricularis” line 192
Line 235: You write "sternocleidomastoid muscle" and you should name it "sternocephalicus" to the mastoid process.
A: In the line 234, “sternocleidomastoid” was replaced to sternocephalicus. Line 205
Line 235 and 236: You write "The buccal branch of the facial nerve passed ventrally to the parotid gland". You should say: "The ventral buccal branch of the facial nerve passed ventrally and rostrally to the parotid gland"
A: The sentence was changed as suggested by the reviewer: “The ventral buccal branch of the facial nerve passed ventrally and rostrally to the parotid gland,...” (lines 205-207).
Line 241: The black arrow points at the dorsal buccal branch of the facial nerve not at the ventral. You have not specify if it is dorsal or ventral. However, in the text you seem to refer to the ventral.
A: In the line 226, the sentence was alterated: “Rostrally we still find the masseter muscle (mm), surrounded dorsally by the dorsal facial nerve (dfn), and caudally by the parotid duct (pd).
And in the line 205 we include mention of the dorsal portion of the facial nerve: “The ventral buccal branch of the facial nerve passed ventrally and rostrally to the parotid gland, and the dorsal portion of the facial nerve passes over the parotid gland (Figure 1).”
Line 242: The same applies to the "sternocleidomastoid muscle"
A: “sternocleidomastoid muscule was replaced at “sternocephalicus” muscule.
Have you study the sympathetic and parasympathetic innervation of the salivary glands?
A: No, we have not yet studied the sympathetic and parasympathetic innervation of the glands, but this could be a topic for future work for this and other species of wild animals that we have, since this study is already very complex and extensive.
The histology images are a bit blur. May be is the quality of the pdf. You should check it.
A: The images were recorded with at least 300 dpi, so I believe it is a problem with the PDF version. I prefer to upload the images in TIFF format to the system, but I have to send them as EPS (format accepted by the system).

Reviewer 2 Report
Comments and Suggestions for Authors
Morphology of Larger Salivary Glands in Peccaries (Pecari tajacu Linnaeus, 1758)
The authors have developed a very complete study on the large salivary glands of the collared peccary, applying various methodologies. I think the article is correct, but it has many gaps to improve. I think that the authors should look for a more applied justification explaining the purpose of the study.
On the other hand, the length of the article is excessive. There is much anecdotal information that has not been discussed, and should be removed or placed in Supplementary Information. Furthermore, I do not see it appropriate to include Results and Discussion together in this study since the difference between both is clear. The separation of the sections will allow the authors to make a joint discussion of the results.
The editing in English must be reviewed
Simple Summary
It should be much more concise in the objectives of the study, methodology and results. The authors have focused on explaining that the differences between peccaries and pigs are important, and that justifies the difference between salivary glands. It is true, but apart from this they have not given further information on results.
Abstract
L26. “This work aims to study the major salivary glands of peccaries during their growth, providing information of interest to the clinics, surgery, nutrition, and conservation of the species.”. I think the authors should be more realistic and indicate the direct benefits of the study. This sentence should be at the end of the Abstract.
Simple Summary and Absrtact are curiously too different, they seem to be different paragraphs of the same large Abstract. I think they should reconsider what information to put in each section.
L40: “These glands have differences when compared to other species, including hogs.”. why do they specify “hog”? This is not an attractive conclusion. The authors mainly show the macroscopic “differences”, but they are similar to those of other domestic species. They should think of a slightly more striking (but realistic) conclusion, and include the usefulness of the study (also realistically). Even, the authors should include some physiological inferences that derives directly from the study.
Introducción
L48: In addition to these, the presence of zygomatica is reported in carnivores and agoutis (Dasyprocta leporina) [1]; and also, molars, in felines [2].”. I find it strange that they specifically mention Dasyprocta leporine. Why?
Why don't you mention the zyogomatic gland with the rest of glands?
L56: “In general, the major salivary glands, together with the minor salivary glands, are responsible for the perfect maintenance of oral physiology”. They then refer to all glands in general? Do they guarantee “perfect maintenance”? The authors should be more precise; I think the “perfect” is a non-appropriated term.
L63: “Despite their importance, the morphology and histochemical characteristics of these glands in wild animals are not widely reported [1,6,7].” The authors should specify which domestic species have been studied in depth, and the example of wild species studied.
L67: “Studies suggest that 67 the salivary glands played a fundamental role in the terrestrial domination of mammals 68 in various niches [8,9], contributing to evolution, as a result of modifying themselves to 69 facilitate the digestion of the most diverse foods.”. Repeated sentence. On the other hand, this sentence should be leading the justification of the study…
L74: “There are reports that salivary glands help species 74 adapt to diets containing antinutritional factors, such as tannin in hogs [4].”. Hogs suddenly appear, but we know that they will be the key species for comparison. Iin the Introduction, the authors should try to make a non-accidental appearance of “hogs”, but rather methodological.
L76-92. This paragraph should be in Discussion, here there is too much detailed information and it does not allow us to understand the justification of the study.
L93-99: The authors must rethink the justification of the study. The fact that the collared Peccary is being bred in captivity does not justify the study, it does allow the biological material to be available, but it is not a conceptual justification. Why peccaries and why salivary glands?
Finally, in the Introduction a sentence should be included explaining the importance of the study. How… “to facilitate the management and preservation of 99 this species”
Material and Methods
2.1. Location and Collection of Material
sex and age distribution?
L105: Delete: “It occupies an area of 20 ha and its measurements georeferencing are between the geographic coordinates 5º 11' S and 37º 20' W Gr.”
L106: If you consider that diet could be an influential factor in secretion, detail the diet in the breeding system.
2.3. Macroscopic Analysis
It is not necessary to describe the access dissection protocol, but yes the features described. Delete the subtitle “Glandular Morphometry”
For further studies, I also recommend measuring volume, which may be related to weight; but allows the calculation of weight/volume relationship (density).
2.4. Microscopic Analysis and Histological Staining
Delete “Histological Staining”
L132: To reduce biases and possible interference, the specimens went through an isotropization process. That is, they went through a process so that all regions of the sample had the same opportunity to be seen under the microscope. The sections were made randomly.”. It would have been more representative to make serial sections. For example, one section every 20 sections. All the initial explanation is not necessary, the important thing is “representativeness”
How many sections were made of each gland?
“Tissue fragments measuring 0.5 cm³ from… “. The size of the tissue is not important, contrary, the thickness of the section. Please include
L138-144: Please be shorter and avoid repetitions.
Here, the authors stated that the major glands are studied, it should be mentioned in the objective.
2.5. Immunohistochemistry
L150. “from four- and seven-month-old specimens were adhered to silanized slides”. It is the only time the age is detailed. Is there any reason?
They should have included a control of a domestic species with known results.
2.6. Transmission Electron Microscopy
L181; “Different portions of the parotid…”.. random? Different? How many per organ?
There is no statistical analysis. The authors should include analyses demonstrating gland growth, and differences between sex and body side.
Results and Discussion
The authors have included in this section information in 24 pages and 17 Figures, which is excessive. Authors should think carefully about distributing images here and on Supplementary materials. They should be much more concise in presenting information. On many occasions, they are including anecdotal information that is not used to make comparisons between species, nor to explain physiological inferences. I would recommend that (including Figures), this section be approximately 12-15 pages.
I recommend a separate presentation of Results and Discussion. The information in both section is very clear. In Discussion, the authors should focus on the main results and develop a good morphological comparison between species, and important functional inferences.
L209-219: This paragraph presents information that in part must be eliminated, and the physiological part included in the Introduction, because justifies the study.
L218: “This species has been the target of commercial farms to generate meat, and thus 217 requires studies that support strategies for the formulation of diets and nutrition, both for 218 commercial production and for the conservation of the species.”. Here there is a more realistic justification. If compared with pigs, they should establish the difference in nutrition between both species. All this should be placed in Introduction.
In the first paragraph, authors should present the main results of the study, but not in detail.
“With a superficial dissection and removal of the parotid auricular muscle,”. Do not talk about the dissection description, but yes the topographic position of the gland. Please, apply this comment when necessary.
L224-227: I am surprised that the comparisons are primarily made with wild animals, and not with domestic ones. Domestic animals should be the reference animals, and specifically focus on the species that are taxonomically closest to the peccary. Please, apply this comment when necessary.
Figure 1 should be used to place topographical references, and abbreviate the text. Uniformize the legend, and use the same terminology, sometimes they use letters, sometimes asterisks and arrows and colors. I recommend using lowercase letters. They should identify more but selected structures. I do not understand the identification criteria, since they do indicate the sternocleidomastoid muscle, but not the masseter muscle.
L251: “An increase in the weight and size (width, thickness, and length) of the parotid gland is observed as the animal ages (Table 2), demonstrating that it is important in the animal's adult life.”. This conclusion requires an statistical analysis. As for “importance”, it is a very simple criterion to say that it is important because it grows.
I suggest transforming Table 2 into Figure 2 allowing the visualization of size changes.
These same suggestions apply to the description of the other glands.
Comments on the Quality of English LanguageThe editing in English must be reviewed
Author Response
Referee II
Morphology of Larger Salivary Glands in Peccaries (Pecari tajacu Linnaeus, 1758)
The authors have developed a very complete study on the large salivary glands of the collared peccary, applying various methodologies. I think the article is correct, but it has many gaps to improve. I think that the authors should look for a more applied justification explaining the purpose of the study.
On the other hand, the length of the article is excessive. There is much anecdotal information that has not been discussed, and should be removed or placed in Supplementary Information. Furthermore, I do not see it appropriate to include Results and Discussion together in this study since the difference between both is clear. The separation of the sections will allow the authors to make a joint discussion of the results.
A: We appreciate the reviewer's considerations and would like to thank him for his other suggestions, which will be taken into consideration. The article is indeed long, and we wrote the results together with the discussion precisely in an attempt to reduce the size of the article. However, we decided not to break the study down into several smaller articles precisely in order to provide the public with a high-quality and complete article with different techniques, despite its size. We chose Plos One for this purpose because of the quality of the articles we have already published in it, as well as other articles from it that we use as references.
The editing in English must be reviewed
A: The translation into English was done by an American citizen named Betty Jean Brandt de Oliveira, however, technical writing errors may appear. Therefore, after requesting proofreaders, we will send the English version to another company to do the proofreading.
Simple Summary
It should be much more concise in the objectives of the study, methodology and results. The authors have focused on explaining that the differences between peccaries and pigs are important, and that justifies the difference between salivary glands. It is true, but apart from this they have not given further information on results.
Abstract
L26. “This work aims to study the major salivary glands of peccaries during their growth, providing information of interest to the clinics, surgery, nutrition, and conservation of the species.”. I think the authors should be more realistic and indicate the direct benefits of the study. This sentence should be at the end of the Abstract.
A: Thank you for your request to change the Abstract. We have reduced the objectives of the Abstract by placing some information at the end of the Abstract.
Line 26: “This work aims to study the major salivary glands morphology of peccaries during their....”
Lines 40-41: “....other species, including hogs, providing information of interest to the clinics, surgery, nutrition, and conservation of the species.”
Simple Summary and Absrtact are curiously too different, they seem to be different paragraphs of the same large Abstract. I think they should reconsider what information to put in each section.
A: We tried to include information about the state of the art of the subject in the Summary, and the results in the Abstract. We put this idea into practice precisely so that Summary and Abstract would not be similar and could complement each other. But we can change it if you really think it is necessary.
L40: “These glands have differences when compared to other species, including hogs.”. why do they specify “hog”? This is not an attractive conclusion. The authors mainly show the macroscopic “differences”, but they are similar to those of other domestic species. They should think of a slightly more striking (but realistic) conclusion, and include the usefulness of the study (also realistically). Even, the authors should include some physiological inferences that derives directly from the study.
A’: We specifically mention pigs because collared peccaries are always compared to pigs, due to their phenotypic similarities, however, both have great anatomical differences, mainly in the digestive system, as we mentioned in the Summary.
A’’: We agree that we need to improve the conclusion of the Abstract, and include something that relates morphology to physiology; we appreciate this request. Therefore, we have changed the conclusion of the Abstract.
“...provides scientific support for a better nutritional strategy for this species, which includes changing from a pig-based diet to one with more fiber, taking advantage of the differences in salivary and stomach anatomy of peccary, helping to reduce the costs of breeding the species, and also helps in the conservation of the species in wildlife screening centers.” (Lines 40-43)
A```: We mainly changed the conclusion of the article as can be seen later.
Introducción
L48: In addition to these, the presence of zygomatica is reported in carnivores and agoutis (Dasyprocta leporina) [1]; and also, molars, in felines [2].”. I find it strange that they specifically mention Dasyprocta leporine. Why?
A: Dasyprocta leporine, is the agouti; we mention this species because it is another Brazilian wild animal that coexists with the collared peccary, and also because it is one of the only species of Brazilian wild animals to have published research regarding its salivary glands.
Why don't you mention the zyogomatic gland with the rest of glands?
A: In fact, this entire sentence does not make sense to be present in the introduction, since both the zygomatic gland and the molar gland are minor salivary glands, and the present study aims to analyze the major salivary glands. Therefore, we removed this sentence. Thank you for the observation.
L56: “In general, the major salivary glands, together with the minor salivary glands, are responsible for the perfect maintenance of oral physiology”. They then refer to all glands in general? Do they guarantee “perfect maintenance”? The authors should be more precise; I think the “perfect” is a non-appropriated term.
A: We agree with the reviewer's recommendation and changed the sentence to: “In general, the major salivary glands, together with the minor salivary glands, contribute to the maintenance of oral physiology.” (line 54-55).
L63: “Despite their importance, the morphology and histochemical characteristics of these glands in wild animals are not widely reported [1,6,7].” The authors should specify which domestic species have been studied in depth, and the example of wild species studied.
A: We prefer not to include in the introduction which species had their major salivary glands studied morphologically and histochemically, since they are mentioned in the results/discussion. However, in terms of domestic species, all have been studied (horses, cows, small ruminants, pigs and minipigs). In terms of studies in wild animals, very few species have these studies completed, such as agouti and ferret.
L67: “Studies suggest that 67 the salivary glands played a fundamental role in the terrestrial domination of mammals 68 in various niches [8,9], contributing to evolution, as a result of modifying themselves to 69 facilitate the digestion of the most diverse foods.”. Repeated sentence. On the other hand, this sentence should be leading the justification of the study…
A: The repeated sentence was removed.
L74: “There are reports that salivary glands help species 74 adapt to diets containing antinutritional factors, such as tannin in hogs [4].”. Hogs suddenly appear, but we know that they will be the key species for comparison. Iin the Introduction, the authors should try to make a non-accidental appearance of “hogs”, but rather methodological.
A: Thank you, you made an important point here, I inserted something about the anatomy of the stomach as inserted in the Summary. The sentence: This domestic species, is comparable to the collared peccary (Pecari tajacu), but has different characteristics, including a stomach that is similar to that of ruminants [11]. Peccaries are distributed from the south of the United States to the Andes, reaching northern Argentina. They are artiodactyl mammals, from the Tayassuidae family, initially described by Linnaeus, in 1758, as belonging to the Sus genus [12]. The stomach of the collared peccary contains a glandular portion, and two blind sacs, one cranioventral and one caudo-dorsal [11], allowing for several items within their menu, including fibrous items [13], which should also cause differences in their salivary glands when compared to hogs. (lines 71-78)
L76-92. This paragraph should be in Discussion, here there is too much detailed information and it does not allow us to understand the justification of the study.
A: As suggested by the referee, all suggested content has been removed from the introduction, making it more concise.
L93-99: The authors must rethink the justification of the study. The fact that the collared Peccary is being bred in captivity does not justify the study, it does allow the biological material to be available, but it is not a conceptual justification. Why peccaries and why salivary glands?
A: We accepted the request and changed the justification for carrying out the study, changing the final paragraph to: “The importance of the salivary glands for digestion, adaptation, and homeostasis, and the anatomical differences in the digestive system of collared peccaries and pigs allows the first to digest even fibrous materials, increasing nutritional possibilities. Studies addressing the morphology of tayassuidaes do not involve information about the morphology of the parotid, mandibular and sublingual glands. These animals are being domesticated and used recently as a protein source in commercial farms in Brazil. Taking all these aspects into consideration, this work aims to fill the gap of information about such glands in collared peccaries at different periods of development, in order to support clinical and surgical knowledge, preservation of this species, and, to support strategies for the formulation of diets and nutrition, which nowadays receives nutrition similar to that of pigs. (Lines 79-88)
Finally, in the Introduction a sentence should be included explaining the importance of the study. How… “to facilitate the management and preservation of 99 this species”
A: We changed the idea of ​​the justification of the study, focusing on the fact that the collared peccary has stomach anatomical differences similar to those of small ruminants, which allow for the addition of more fibrous elements to the diet. Therefore, they must also have different salivary glands, which would lead to a different diet from that given to pigs. Therefore, it would be possible to develop better management strategies for this species with a view to animal production.
Material and Methods
2.1. Location and Collection of Material
sex and age distribution?
A: “The work used eight peccaries of different ages (two 4 months old, two 5 months old, two 6 months old, and two 07 months old) of both sexes,....” (lines91-92)
L105: Delete: “It occupies an area of 20 ha and its measurements georeferencing are between the geographic coordinates 5º 11' S and 37º 20' W Gr.”
A: The text was removed as suggested.
L106: If you consider that diet could be an influential factor in secretion, detail the diet in the breeding system.
A: After the request, we insert the following text: “...the diet was grounded for pigs, with 18% protein based on wheat bran, corn and soybean. In addition, once a week complementary feeding was given with fruits, potatoes, carrots and pumpkins..” (lines 94-97)
2.3. Macroscopic Analysis
It is not necessary to describe the access dissection protocol, but yes the features described. Delete the subtitle “Glandular Morphometry”
A: Both the subtitle “Glandula Morphometry” and the description of the dissection access have been removed from the text.
For further studies, I also recommend measuring volume, which may be related to weight; but allows the calculation of weight/volume relationship (density).
A: Thanks for the tip.
2.4. Microscopic Analysis and Histological Staining
Delete “Histological Staining”
A: It was removed.
L132: To reduce biases and possible interference, the specimens went through an isotropization process. That is, they went through a process so that all regions of the sample had the same opportunity to be seen under the microscope. The sections were made randomly.”. It would have been more representative to make serial sections. For example, one section every 20 sections. All the initial explanation is not necessary, the important thing is “representativeness”
A: The text: “To reduce biases and possible interference, the specimens went through an isotropization process. That is, they went through a process so that all regions of the sample had the same opportunity to be seen under the microscope.” was removed.
How many sections were made of each gland?
A: “The slides with 05-μm-thick sections, 08 of each gland per animal, from 4, 5, 6, and 7month old specimens were stained with Hematoxylin-Eosin (HE);...” (lines 119-120)
“Tissue fragments measuring 0.5 cm³ from… “. The size of the tissue is not important, contrary, the thickness of the section. Please include
A: “The slides with 05-μm-thick sections, 08 of each gland per animal, from 04, 05, 06, and 07-old specimens were stained with Hematoxylin-Eosin (HE);....” (lines 119-120)
L138-144: Please be shorter and avoid repetitions.
A: We were unable to summarize this part of the text.
Here, the authors stated that the major glands are studied, it should be mentioned in the objective.
A: “After fixation, the material (major salivary glands) was washed in running water and subjected to the methodology....” (line 117)
2.5. Immunohistochemistry
L150. “from four- and seven-month-old specimens were adhered to silanized slides”. It is the only time the age is detailed. Is there any reason?
A: There was an error in not mentioning the months of collection (4, 5, 6 and 7 months of age) for the histological and histochemical staining methodology and also in placing them incompletely in the immunohistochemistry methodology. The arranged sentences are below:
“The slides with 05-μm-thick sections, 08 of each gland per animal, from 4, 5, 6, and 7month old specimens were stained with Hematoxylin-Eosin (HE);..” (lines 119)
“The same biological samples used for the histology and histochemistry techniques were used in immunohistochemistry. 5-μm-thick sections of the parotid, mandibular, and sublingual glands from from 04, 05, 06, and 07-old specimens were adhered....” (130-132)
They should have included a control of a domestic species with known results.
A: We used the negative control in section without the primary antibody.
2.6. Transmission Electron Microscopy
L181; “Different portions of the parotid…”.. random? Different? How many per organ?
A: “.Three random portions of the parotid, mandibular, and sublingual major salivary glands were....” (Line 162)
There is no statistical analysis. The authors should include analyses demonstrating gland growth, and differences between sex and body side.
A: Unfortunately, we are unable to perform statistics on the size of the salivary glands. Since the animals in this study are from a wild species that will serve as breeding stock for the scientific and conservationist breeding facility at our University (UFERSA), we cannot euthanize a sufficient number to perform statistics within each age group (4, 5, 6 and 7 months). The ideal number would be 8 animals for each sex and age group. In the coming years, with the increase in the herd, after having a sufficient population and reintroducing the initial number of animals into the wild, we will be able to conduct research with more euthanasia.
Results and Discussion
The authors have included in this section information in 24 pages and 17 Figures, which is excessive. Authors should think carefully about distributing images here and on Supplementary materials. They should be much more concise in presenting information. On many occasions, they are including anecdotal information that is not used to make comparisons between species, nor to explain physiological inferences. I would recommend that (including Figures), this section be approximately 12-15 pages.
I recommend a separate presentation of Results and Discussion. The information in both section is very clear. In Discussion, the authors should focus on the main results and develop a good morphological comparison between species, and important functional inferences.
We appreciate the referee's request, however we would like to argue the possibility of not accepting these two requests. When we sent this study to Animals, the idea was precisely to publish a comprehensive study, the largest in the literature for salivary glands for any species, serving to be referenced by any other scientific work involving salivary glands regardless of the species. To this end, several techniques were used for each major salivary gland (histological staining, histochemistry, immunohistochemistry, transmission electron microscopy, scanning electron microscopy, topography and biometry). This way, there would be no way for us to reduce the text or the figures without removing some of these techniques or fragmenting the work by gland, and the idea is exactly not to subdivide the article, unless the Journal wants to do so. We are against scientific fragmentation. Given the size of the article, we prefer to leave results and discussion together, since separating the two will make the text even longer. If it is possible to keep it that way, we prefer it.
L209-219: This paragraph presents information that in part must be eliminated, and the physiological part included in the Introduction, because justifies the study.
A: Request accepted, we removed these sentences from the results/discussion and supplemented the introduction, giving more robustness to the study's justification. The sentence added to the introduction was:
“This domestic species, is comparable to the collared peccary (Pecari tajacu), but has different characteristics, including a stomach that is similar to that of ruminants [11]. Peccaries are distributed from the south of the United States to the Andes, reaching northern Argentina. They are artiodactyl mammals, from the Tayassuidae family, initially described by Linnaeus, in 1758, as belonging to the Sus genus [12]. The stomach of the collared peccary contains a glandular portion, and two blind sacs, one cranioventral and one caudo-dorsal [11], allowing for several items within their menu, including fibrous items [13], which should also cause differences in their salivary glands when compared to hogs.
The importance of the salivary glands for digestion, adaptation, and homeostasis, and the anatomical differences in the digestive system of collared peccaries and pigs allows the first to digest even fibrous materials, increasing nutritional possibilities. Studies addressing the morphology of tayassuidaes do not involve information about the morphology of the parotid, mandibular and sublingual glands. These animals are being domesticated and used recently as a protein source in commercial farms in Brazil. Taking all these aspects into consideration, this work aims to fill the gap of information about such glands in collared peccaries at different periods of development, in order to support clinical and surgical knowledge, preservation of this species, and, to support strategies for the formulation of diets and nutrition, which nowadays receives nutrition similar to that of pigs. “ (Lines 71-88)
L218: “This species has been the target of commercial farms to generate meat, and thus 217 requires studies that support strategies for the formulation of diets and nutrition, both for 218 commercial production and for the conservation of the species.”. Here there is a more realistic justification. If compared with pigs, they should establish the difference in nutrition between both species. All this should be placed in Introduction.
A: Thank you for the suggestion, it was accepted, so the last paragraph of the Introduction was changed:
The importance of the salivary glands for digestion, adaptation, and homeostasis, and the anatomical differences in the digestive system of collared peccaries and pigs allows the first to digest even fibrous materials, increasing nutritional possibilities. Studies addressing the morphology of tayassuidaes do not involve information about the morphology of the parotid, mandibular and sublingual glands. These animals are being domesticated and used recently as a protein source in commercial farms in Brazil. Taking all these aspects into consideration, this work aims to fill the gap of information about such glands in collared peccaries at different periods of development, in order to support clinical and surgical knowledge, preservation of this species, and, to support strategies for the formulation of diets and nutrition, which nowadays receives nutrition similar to that of pigs. (line 88)
In the first paragraph, authors should present the main results of the study, but not in detail.
“With a superficial dissection and removal of the parotid auricular muscle,”. Do not talk about the dissection description, but yes the topographic position of the gland. Please, apply this comment when necessary.
A: With a superficial dissection and removal of the parotidoauricularis muscle, it was possible to observe the parotid gland (one on each side), which is the largest salivary gland, completely covering the mandibular gland, being located caudal to the masseter muscle. (Lines 192-194)
L224-227: I am surprised that the comparisons are primarily made with wild animals, and not with domestic ones. Domestic animals should be the reference animals, and specifically focus on the species that are taxonomically closest to the peccary. Please, apply this comment when necessary.
A: In fact, we use comparisons with wild and domestic animals. The idea of ​​making comparisons with other wild animals would be because there are few studies that deal with the salivary glands of wild animals, and this article could serve as a reference for others to come. Furthermore, there are few studies with Brazilian wild animals and it is only now that the use of Brazilian wild animals in commercial breeding is beginning due to bureaucratic obstacles, requiring all types of studies, from anatomical, physiological or health of these species. When making comparisons with domestic animals, we prefer to do so with pigs, since they are the most similar species, despite both species having anatomical differences in their digestive system and in the functionality of the major salivary glands, as demonstrated in the present study.
Figure 1 should be used to place topographical references, and abbreviate the text. Uniformize the legend, and use the same terminology, sometimes they use letters, sometimes asterisks and arrows and colors. I recommend using lowercase letters. They should identify more but selected structures. I do not understand the identification criteria, since they do indicate the sternocleidomastoid muscle, but not the masseter muscle.
A: We appreciate the suggestion, we changed the identifications, including lowercase letters in the identifications and increased some identifications.
“Figure 1. Photographic image of the macroscopy of the major salivary glands of the collared peccary (Pecari tajacu Linnaeus, 1758). Mandibular gland (mg) is located ventrally to the linguofacial vein (lv). Above is part of the parotid gland (pg), which is rostrally to the sternocephalicus muscle (em). The “em” is positioned dorsally to the thymus (t). Rostrally we still find the masseter muscle (mm), surrounded dorsally by the dorsal facial nerve (dfn), and caudally by the parotid duct (pd).
(lines 223-227).
Figure 1 after alterations:
L251: “An increase in the weight and size (width, thickness, and length) of the parotid gland is observed as the animal ages (Table 2), demonstrating that it is important in the animal's adult life.”. This conclusion requires an statistical analysis. As for “importance”, it is a very simple criterion to say that it is important because it grows.
A: As previously stated, we do not have a sufficient sample number to make statistics, as these are wild animals with a small population of individuals and which will still be used as breeders, and cannot currently carry out further euthanasia. The text was changed to avoid giving the idea that there was statistical significance, and also to minimize the statement of a possible importance of the parotid salivary gland in adults:
“An increase in the weight and size (width, thickness, and length), at least numerically, of the parotid gland is observed as the animal ages (Table 2), demonstrating that the function of this salivary gland could be important in the animal's adult life.” (lines 236-238).
The same was done in the text that mentions the mandibular gland:
“Regarding the variation in mandibular weight, it was found that it decreases as the animal grows, while the width, thickness, and length of the gland increases as the animal ages (Table 3), at least numerically, like in hogs [4]. This gland could be more important during the fetal period.” (lines 274).
And for sublingual salivary glands:
“The morphometric analysis of the polystomatic portion demonstrated that the weight, width, thickness, and length of the gland increased with the advancing age of the animal, at least numerically, and the right and left antimeres showed divergences (Table 4).” (lines 298)
“The morphometric analysis of the monostomatic portion demonstrated that its weight, width, thickness, and length of the gland increased with the advancing age of the animal, at least numerically. Furthermore, the right and left antimeres also showed divergences (Table 5).” (Line 309)
I suggest transforming Table 2 into Figure 2 allowing the visualization of size changes.
These same suggestions apply to the description of the other glands.
A: We believe it is important to keep the figures with the photomicrographs of the histological characterization of the glands. The figures with only the sizes of the glands would be more appropriate if the work had little material to be presented, but we already have a lot.
We are immensely grateful to the present referee who, with his requests, greatly improved the quality of this article. We tried to comply with all the suggestions requested, however, unfortunately, some of them we still cannot do due to some type of impediment.
We improved the end of the discussion:
Analyzing all the results obtained in the present study, we can state that the parotid in peccary is much smaller when compared to that of pigs, even resembling the one present in domestic ruminants, and has a different transaction as well, producing more acidic mucins than in pigs (positive for Alcian Blue, while in pigs it is negative). The mandibular gland has a greater production of acidic and basic mucopolysaccharides than the parotid gland; and the sublingual, greater production than the mandibular itself, demonstrated by the high positivity to PAS, AB and PAS+AB, in addition to the latter also being very positive to the lecithins studied. Such productions are also higher than in pigs [22].
Such different salivary characteristics from those discovered in pigs, combined with the differentiated stomach anatomy, with four compartments (gastric pouch, cranioventral blind sac, caudodorsal blind sac and direct compartment), similar to those of ruminants [11,12], lead these animals to be capable of to digest more fibrous foods more efficiently through fermentation in these compartments. Therefore, it is necessary to readjust the nutrition of this species, moving from a diet based on pig feed to one that includes more fibrous foods, which could even reduce the costs of raising these animals, while increasing production. (Lines 697-712)
Completion has also been improved: This is the most complete article in literature concerning the morphology of salivary glands in any species of wild animal. The collared peccary has parotid salivary glands with serous secretion, and mandibular and sublingual glands with mucous secretion, differing from other species, including hogs. These differences were also observed in the type of mucin secretion due to PAS and Alcian Blue positivity. It was observed that the glands change their weight, size, and type of secretion, with development, even decreasing in weight, as is the case of the collared peccary gland, or having morphometric differences related to the side of the gland, as in the case of sublingual glands. This study, including positivity for some types of lecithin and electron microscopy, provides scientific support for a better nutritional strategy for this species, which includes changing from a pig-based diet to one with more fiber, taking advantage of the differences in salivary and stomach anatomy of peccary, helping to reduce the costs of breeding the species, and also helps in the conservation of the species in wildlife screening centers. (Lines 714-726)
A: Bibliographic references were reduced from 57 to 50.

Round 2
Reviewer 2 Report
Comments and Suggestions for Authors
Morphology of Larger Salivary Glands in Peccaries 2 (Pecari tajacu Linnaeus, 1758)
Thank you for submitting your review of the manuscript. I still think that the draft is excessively long, with a lot of anecdotal information. I understand the authors' position, because it is always difficult for us to give up information; but it is essential to do synthesis exercises. I leave this consideration to the assistant editor. My opinion is clear in this regard: The article should be considerably reduced in length and the number of figures should be rigorously selected.
Specific comments:
Simple Summary
I really think that Simple Summary and Abstract are not two paragraphs that build the same section, they are different sections. I understand that both must explain objectives, methods, results and conclusions from different perspectives. But Simple Summary is not a background.
Abstract
Last Line: “…helping to improve the welfare of the species in wildlife captive centers.”
Question from first revision:
L40: “These glands have differences when compared to other species, including hogs.”. why do they specify “hog”? This is not an attractive conclusion. The authors mainly show the macroscopic “differences”, but they are similar to those of other domestic species. They should think of a slightly more striking (but realistic) conclusion, and include the usefulness of the study (also realistically). Even, the authors should include some physiological inferences that derives directly from the study.
A’: We specifically mention pigs because collared peccaries are always compared to pigs, due to their phenotypic similarities, however, both have great anatomical differences, mainly in the digestive system, as we mentioned in the Summary.
R2: Ok, then, briefly include such differences.
I think that the criteria for selecting species for comparisons are not consistent. I think they should be more consistent with the available information. I understand that there must be two essential criteria: 1) Available information, and 2) phylogenetic relationships.
Question from first revision:
L63: “Despite their importance, the morphology and histochemical characteristics of these glands in wild animals are not widely reported [1,6,7].” The authors should specify which domestic species have been studied in depth, and the example of wild species studied.
A: We prefer not to include in the introduction which species had their major salivary glands studied morphologically and histochemically, since they are mentioned in the results/discussion. However, in terms of domestic species, all have been studied (horses, cows, small ruminants, pigs and mini pigs). In terms of studies in wild animals, very few species have these studies completed, such as agouti and ferret.
R2: The authors should say that it is a topic well studied in domestic animals and scarcely studied in wild animals.
L81-83. “The importance of the salivary glands for digestion, adaptation, and homeostasis, and the anatomical differences in the digestive system of collared peccaries and pigs allows the first to digest even fibrous materials, increasing nutritional possibilities.” If this sentence has been scientifically proven (in peccaries), please include references. Meanwhile I think it's a hypothesis. Then, two possibilities: A) Explain the hypothesis, clarifying that it is a hypothesis, or B) Include that reference about pigs (or in general), with a bibliographical citation.
Please standardize the use of “pigs” or “hogs”
Objective of the study: Why studying the salivary glands in peccaries? Why the salivary glands?; why in this species (peccaries)? What makes it scientifically sounding? The argument of “scarce studies” does not convince me. I think the authors should think again to make the article more attractive and useful. The dilemma is that peccaries are frugivorous animals with a stomach with various compartments, more or less similar to ruminants? In their response, the authors mentioned the objective more clearly and forcefully: “We changed the idea of the justification of the study, focusing on the fact that the collared peccary has stomach anatomical differences similar to those of small ruminants, which allow for the addition of more fibrous elements to the diet. Therefore, they must also have different salivary glands, which would lead to a different diet from that given to pigs. Therefore, it would be possible to develop better management strategies for this species with a view to animal production.”. I think this explanation should be improved.
Could the authors clearly mention which is the Introduction sentence that expresses the objective of the study?
L 93: “The study used eight peccaries, including two individuals (male and female) at four different ages (4, 5, 6 and 7 months).”. So the study doesn't include any adult individuals? It is a major weakness of the study. This must be considered in Discussion section.
L 96-99. The administration of a pig nutrition could have consequences with variations in the salivary glands of peccaries? For example, it is known that diet is essential for the development of the stomach. I think it is necessary to consider it in Discussion.
L 119-121: “After fixation, the major salivary glands were washed in running water and subjected to the methodology adapted from Tolosa et al. [18] to obtain the slides. The slides 120 with 05-μm-thick sections, 08 of each gland per animal”. Delete “, from 4, 5, 6, and 7month old specimens”. If the age have been already mentioned above, it is no longer necessary. Delete ages also in L 134.
2.5. Immunohistochemistry
If authors have used a negative control, they must specify this.
I understand the limitation to increasing sample size, which I would not support either. But considering this restriction, the authors must be extremely careful when using the terms “Major” or “Minor”, ​​“More” or “Less”, because in science they can only be mentioned when there is statistical support. So I recommend the authors to mention this limitation in Discussion. The expression “at least numerically” is not correct. Making an appropriate and previous explanation is the key in my opinion.
Results and Discussion
In my opinion, this is the great flaw of this study. It is a difficult manuscript to follow due to the inability to highlight the most important issues. The authors must considerably improve the attractiveness of their study. My previous proposals were two: A) Synthesize and B) Separate sections. The authors believe that neither is appropriate. I could be flexible with B), but I strongly disagree with A). The authors must make a considerable effort to synthesize. The authors show serious problems in synthesizing, but it is necessary. In addition, most graphic material could be presented as supplementary figures. All researchers and all the journals make this effort. I don't think there should be exceptions.
Finally, morphological comparisons with wild and domestic species go beyond their wild/domestic status. Phylogenetic comparisons and comparisons with species where there is better information should be chosen. The fact that peccaries and agoutis are wild is not enough of an argument to prioritize this comparison.
Author Response
Referee’s comments
Thank you for submitting your review of the manuscript. I still think that the draft is excessively long, with a lot of anecdotal information. I understand the authors' position, because it is always difficult for us to give up information; but it is essential to do synthesis exercises. I leave this consideration to the assistant editor. My opinion is clear in this regard: The article should be considerably reduced in length and the number of figures should be rigorously selected.
A: To reduce the length of the article, we have removed figures 7, 8, 9, 10, 11, 12, 13, 14, 15, 16 and 17, which will be made available as supplementary material (Figure S1-S11).
Specific comments:
Simple Summary
I really think that Simple Summary and Abstract are not two paragraphs that build the same section, they are different sections. I understand that both must explain objectives, methods, results and conclusions from different perspectives. But Simple Summary is not a background.
A: Thank you for insisting on changing the content of the Simple Summary, we believe it looks much better now: “Simple Summary: Peccaries are distributed from the southern United States to southern South America. Although they were initially described as being from the pig genus Sus, they are very different. This species is recently being used on commercial farms as a source of animal protein, however, there are no studies about their salivary glands, which are so important for digestion and for supporting feeding strategies. Thus, this work aims to study the major salivary glands morphology of peccaries during their growth. During the growth, parotid enlarges and mandibular gland loses weight. Histologically, the parotid has serous production, and sublingual has mucous production, resembles most species, however, mandibular gland produces mucous, unlike other animals, including pigs, which produce seromucous secretion. Histochemically, parotid produces more acidic mucins than pigs and it undergoes maturation during development; mandibular, and especially the sublingual gland, produce more acidic and basic mucopolysaccharides than pigs. The major salivary glands were positive to different lecithins, which were also more positive than in pigs. We conclude that peccaries have a salivary secretion that facilitates the digestion of carbohydrates, improving digestibility and performance; and biometric characteristics and positivity to lecithins that facilitate adaptation to foods with antinutritional factors, being a promising production animal.” (Lines 16-32)
Abstract
Last Line: “…helping to improve the welfare of the species in wildlife captive centers.”
Question from first revision:
L40: “These glands have differences when compared to other species, including hogs.”. why do they specify “hog”? This is not an attractive conclusion. The authors mainly show the macroscopic “differences”, but they are similar to those of other domestic species. They should think of a slightly more striking (but realistic) conclusion, and include the usefulness of the study (also realistically). Even, the authors should include some physiological inferences that derives directly from the study.
A’: We specifically mention pigs because collared peccaries are always compared to pigs, due to their phenotypic similarities, however, both have great anatomical differences, mainly in the digestive system, as we mentioned in the Summary.
R2: Ok, then, briefly include such differences.
A: We also appreciate you requesting changes to the Abstract, which is completely changed: “This work aims to study the major salivary glands morphology of peccaries during their growth. The glands were analyzed by macroscopic description, light microscopy, electron microscopy, histochemistry, and immunohistochemistry. Topographically the salivary glands resemble other animals, including domestic animals and pigs. During the growth, parotid enlarges and mandibular gland loses weight. Histologically, the parotid has serous production, and sublingual has mucous production, resembles most species, however, mandibular gland produces mucous, unlike other animals, including pigs, which produce seromucous secretion. Histochemically, parotid produces more acidic mucins than pigs and it undergoes maturation during development; mandibular, and especially the sublingual gland, produce more acidic and basic mucopolysaccharides than pigs. The results found with transmission and scanning electron microscopy techniques corroborate the histological and histochemistry findings. The major salivary glands were positive to different lecithins (Com-A, BSA-I-B4, WGA and PNA), which were also more positive than in pigs and sheep. We conclude that collared peccaries have a salivary secretion that facilitates the digestion of carbohydrates, and biometric characteristics and positivity to lecithins that facilitate adaptation to foods with antinutritional factors.” (Lines 33-48)
I think that the criteria for selecting species for comparisons are not consistent. I think they should be more consistent with the available information. I understand that there must be two essential criteria: 1) Available information, and 2) phylogenetic relationships.
Question from first revision:
L63: “Despite their importance, the morphology and histochemical characteristics of these glands in wild animals are not widely reported [1,6,7].” The authors should specify which domestic species have been studied in depth, and the example of wild species studied.
A: We prefer not to include in the introduction which species had their major salivary glands studied morphologically and histochemically, since they are mentioned in the results/discussion. However, in terms of domestic species, all have been studied (horses, cows, small ruminants, pigs and mini pigs). In terms of studies in wild animals, very few species have these studies completed, such as agouti and ferret.
R2: The authors should say that it is a topic well studied in domestic animals and scarcely studied in wild animals.
A: Following the suggestion, the sentence was rewritten: “Despite their importance, the morphology and histochemical characteristics of these glands are restricted to domestic animals such as pigs, sheep, cattle and horses, and practically non-existent in wild animals [4,5,6].” (lines 67-70)
L81-83. “The importance of the salivary glands for digestion, adaptation, and homeostasis, and the anatomical differences in the digestive system of collared peccaries and pigs allows the first to digest even fibrous materials, increasing nutritional possibilities.” If this sentence has been scientifically proven (in peccaries), please include references. Meanwhile I think it's a hypothesis. Then, two possibilities: A) Explain the hypothesis, clarifying that it is a hypothesis, or B) Include that reference about pigs (or in general), with a bibliographical citation.
A: Following the referee's considerations, we have removed the following content from the Introduction: “..which should also cause differences in their salivary glands when compared to pigs.”; and “The importance of the salivary glands for digestion, adaptation, and homeostasis, and the anatomical differences in the digestive system of collared peccaries and pigs allows the first to digest even fibrous materials, increasing nutritional possibilities.”
Therefore, the last paragraph of the Introduction was as follows: “Studies addressing the morphology of tayassuidaes do not involve information about parotid, mandibular and sublingual glands, which have great importance for digestion, adaptation, and homeostasis. These animals are being domesticated and used recently as a protein source in commercial farms in Brazil. Therefore, this work aims to fill the gap of information about morphology of larger salivary glands in collared peccaries at different periods of development, generating subsidies to the preservation of this species in wildlife screening centers, and to develop strategies for the formulation of diets and nutrition.” (Lines 86-92)
Please standardize the use of “pigs” or “hogs”
A: We standardize de text with “pigs”.
Objective of the study: Why studying the salivary glands in peccaries? Why the salivary glands?; why in this species (peccaries)? What makes it scientifically sounding? The argument of “scarce studies” does not convince me. I think the authors should think again to make the article more attractive and useful. The dilemma is that peccaries are frugivorous animals with a stomach with various compartments, more or less similar to ruminants? In their response, the authors mentioned the objective more clearly and forcefully: “We changed the idea of the justification of the study, focusing on the fact that the collared peccary has stomach anatomical differences similar to those of small ruminants, which allow for the addition of more fibrous elements to the diet. Therefore, they must also have different salivary glands, which would lead to a different diet from that given to pigs. Therefore, it would be possible to develop better management strategies for this species with a view to animal production.”. I think this explanation should be improved.
A: The objective of the study was changed. The work aims to describe the morphology of the major salivary glands of the collared peccary because there are still no studies on the glands of these animals. There is also another species of Tayassuidae, which we also have in our Center for the Proliferation of Wild Species. This other species also has no studies covering its salivary glands (the next study will be with this other species). The idea of ​​working with the salivary glands is because they are very important for digestion and adaptation of animals to different diets. In addition, only now are commercial breeding sites for the production of collared peccary meat being authorized by the authorities. Before, it was prohibited, so there were really no studies on the species. Another important organ for digestion is the stomach, and the stomach of the collared peccary has already been studied, and it really does have different characteristics from those of pigs, and is more similar to that of ruminants. Despite this, we reduced the information on the stomach, because it really seemed that the differences in the stomach of the collared peccary were part of the objective of the work.
Could the authors clearly mention which is the Introduction sentence that expresses the objective of the study?
A: “Therefore, this work aims to fill the gap of information about morphology of larger salivary glands in collared peccaries at different periods of development, generating subsidies to the preservation of this species in wildlife screening centers, and to develop strategies for the formulation of diets and nutrition.” (Line 86-92)
L 93: “The study used eight peccaries, including two individuals (male and female) at four different ages (4, 5, 6 and 7 months).”. So the study doesn't include any adult individuals? It is a major weakness of the study. This must be considered in Discussion section.
A: The peccary are precocious, they can reproduce at 8 months of age, with an individual at 7 months already mature in relation to its digestive system. Unlike pigs that are born completely defenseless and in large litters, peccaries are usually the birth of two cubs that run after their mother a few hours after birth.
L 96-99. The administration of a pig nutrition could have consequences with variations in the salivary glands of peccaries? For example, it is known that diet is essential for the development of the stomach. I think it is necessary to consider it in Discussion.
A: We believe not, we believe that the administration of pig feed has a greater effect on the intestinal villi, increasing them, but not on the salivary glands.
L 119-121: “After fixation, the major salivary glands were washed in running water and subjected to the methodology adapted from Tolosa et al. [18] to obtain the slides. The slides 120 with 05-μm-thick sections, 08 of each gland per animal”. Delete “, from 4, 5, 6, and 7month old specimens”. If the age have been already mentioned above, it is no longer necessary. Delete ages also in L 134.
A: Suggestion accepted.
2.5. Immunohistochemistry
If authors have used a negative control, they must specify this.
A: The use of a negative control has already been mentioned in the article:
“The negative control was made by using PBS in a humidity chamber instead of lecithin for each tissue.” (lines 148)
In this case, we used histological slides with two tissue sections. In one of them we placed the antibody + PBS, in the other section (negative control) we placed only PBS.
I understand the limitation to increasing sample size, which I would not support either. But considering this restriction, the authors must be extremely careful when using the terms “Major” or “Minor”, ​​“More” or “Less”, because in science they can only be mentioned when there is statistical support. So I recommend the authors to mention this limitation in Discussion. The expression “at least numerically” is not correct. Making an appropriate and previous explanation is the key in my opinion.
A: “In the present study unfortunately, we are unable to perform statistics on the size of the salivary glands. Since the animals in this study are from a wild species that will serve as breeding stock for the scientific and conservationist breeding facility at our University, we cannot euthanize a sufficient number to perform statistics within each age group (4, 5, 6 and 7 months). Therefore, the numbers demonstrate a trend.” (lines 230-239)
A`: We removed the expression. “at least numerically”.
A``: In the methodology for immunohistochemistry, we also added the form of analysis of the slides by the pathologist, emphasizing that the methodology applied is qualitative, a methodology normally applied. “The slide readings were performed by the same pathologist, analyzing the positivity for each antibody by the intensity of the DAB chromogen staining, in a qualitative manner, a technique commonly used in pathology.” (lines 156-158)
Results and Discussion
In my opinion, this is the great flaw of this study. It is a difficult manuscript to follow due to the inability to highlight the most important issues. The authors must considerably improve the attractiveness of their study. My previous proposals were two: A) Synthesize and B) Separate sections. The authors believe that neither is appropriate. I could be flexible with B), but I strongly disagree with A). The authors must make a considerable effort to synthesize. The authors show serious problems in synthesizing, but it is necessary. In addition, most graphic material could be presented as supplementary figures. All researchers and all the journals make this effort. I don't think there should be exceptions.
Finally, morphological comparisons with wild and domestic species go beyond their wild/domestic status. Phylogenetic comparisons and comparisons with species where there is better information should be chosen. The fact that peccaries and agoutis are wild is not enough of an argument to prioritize this comparison.
A: We understand the suggestions and try to demonstrate the main results in the discussion, making the article more attractive. To this end, we changed the entire final text of the discussion, removing explanations that involved the collared peccary's stomach. Such information about the stomach only appears to corroborate the results that demonstrated the greater digestibility of fruits and nuts by peccary compared to pigs, and its suitability for animal production. Therefore, a possibly promising species for animal production as it has characteristics in the salivary glands that allow greater digestibility of non-structural carbohydrates, and characteristics in the stomach that allow greater digestibility of structural carbohydrates.“ Although topographically the salivary glands of peccary resemble other animals, including domestic animals and pigs, some morphological differences were observed. The peccary' parotid enlarges with age, which may be related to the high quantity of foods rich in anti-nutritional factors that these animals find and consume in the forests of South America, as would be fatal for domestic species. This statement is corroborated by the increase in the parotids of pigs when they are subjected to diets rich in antinutritional factors, such as tannin [2]. The gland produces the digestive enzyme amylase, indicating the need to digest carbohydrates, since these animals also feed on fruits and nuts rich in carbohydrates [17]. Decrease in the mandibular gland is also seen in pigs subjected to antinutritional factors [2]. During the growth of mandibular peccaries, this gland loses weight, which increases the possibility that peccaries will adapt to this type of diet with many antinutritional factors. Histologically, the parotid, with serous production, resembles most species, including domestic animals, the same happens for the sublingual gland with mucous production. However, the mandibular gland produces mucous in peccary, unlike other animals, including pigs, which produce seromucous secretion. Such differences in the mandibular glands may also be due to the fact that this gland, unlike the parotid gland that produces saliva when stimulated, is important during sleep [19]. Since peccaries sleep little and are often seen feeding at night [10,11], this gland may have its weight reduced with age, as already mentioned. Still regarding the secretion of the glands, it was observed that the parotid gland produces more acidic mucins than pigs (positive for Alcian Blue, while in pigs it is negative) and that it undergoes maturation during development; and that the mandibular glands, and especially the sublingual gland, produce more acidic and basic mucopolysaccharides than pigs (demonstrated by the high positivity to PAS, AB and PAS+AB). These characteristics provide peccary with better digestion of fruits and nuts when compared to pigs, most similar animal domestic, thus improving the digestibility of food even if nutritionally poorer.
The results found with transmission and scanning electron microscopy techniques corroborate the histological findings, and this is the first study with these techniques in collared peccary salivary glands. This is also the first study to look at positive lecithins in the salivary glands of collared peccaries, which were also more positive than in pigs and sheep [2,16], which also facilitates carbohydrate digestion in collared peccaries. The higher lecithin-positivity also indicates an adaptation of the collared peccaries to the ingestion of foods with antinutritional factors, since pigs when subjected to these factors have increased lecithin-positivity in the salivary glands [2].
In addition to the morphological characteristics mentioned in the larger salivary glands of collared peccaries that provide digestive advantages for carbohydrates when compared to pigs, they also have another advantage, the presence of a stomach with four compartments (gastric pouch, cranioventral blind sac, caudodorsal blind sac and direct compartment), similar to those of ruminants [11,12], lead these animals to be capable of to digest more fibrous foods more efficiently through fermentation in these compartments. Remembering that dietary fibre was once considered to be a negative factor for monogastric animals due to its potential adverse effects on digestibility and performance, but collared peccaries cannot be considered monogastric. Such characteristics point to the collared peccary as a promising production animal, as it can satisfactorily digest both structural and non-structural carbohydrates.” (Lines 606-652)
This change also made it possible to improve the Summary and Abstract.This change also made it possible to improve the Conclusions:This is the most complete article in literature concerning the morphology of salivary glands in any species of wild animal. The collared peccary has a major salivary glands secretion that facilitates the digestion of carbohydrates, improving digestibility and performance; and biometric characteristics and positivity to lecithins that facilitate adaptation to foods with antinutritional factors, being a promising production animal. These results serve as subsidies for the development of diet and nutrition formulation strategies for the species in commercial breeding or in wildlife screening centers. (Lines 654-660) We also accepted the suggestions to leave only the macroscopy and histology figures, leaving the others as figures as supplementary material, thus reducing the article considerably.
A: Comparisons with Tayassu pecari were made because they also belong to the Tayassuidae family, and because they inhabit the same ecological niche as peccary. The agoutis are also part of the same ecological niche. Both compete for food in the same environment.
